EMBO
Molecular Medicine

# SATB2 drives glioblastoma growth by recruiting CBP to promote FOXM1 expression in glioma stem cells

Weiwei Tao[1], Aili Zhang[1], Kui Zhai[1], Zhi Huang[1], Haidong Huang[1], Wenchao Zhou[1], Qian Huang[1], Xiaoguang Fang[1] (iD), Briana C Prager[2,3], Xiuxing Wang[2], Qiulian Wu[2], Andrew E Sloan[4,5], Manmeet S Ahluwalia[6], Justin D Lathia[5,6,7], Jennifer S Yu[1,5,8,9], Jeremy N Rich[2] & Shideng Bao[1,5,8,*] (iD)

## Abstract

Nuclear matrix-associated proteins (NMPs) play critical roles in regulating chromatin organization and gene transcription by binding to the matrix attachment regions (MARs) of DNA. However, the functional significance of NMPs in glioblastoma (GBM) progression remains unclear. Here, we show that the Special AT-rich Binding Protein-2 (SATB2), one of crucial NMPs, recruits histone acetyltransferase CBP to promote the FOXM1-mediated cell proliferation and tumor growth of GBM. SATB2 is preferentially expressed by glioma stem cells (GSCs) in GBM. Disrupting SATB2 markedly inhibited GSC proliferation and GBM malignant growth by downregulating expression of key genes involved in cell proliferation program. SATB2 activates *FOXM1* expression to promote GSC proliferation through binding to the MAR sequence of *FOXM1* gene locus and recruiting CBP to the MAR. Importantly, pharmacological inhibition of SATB2/CBP transcriptional activity by the CBP inhibitor C646 suppressed GSC proliferation *in vitro* and GBM growth *in vivo*. Our study uncovers a crucial role of the SATB2/CBP-mediated transcriptional regulation in GBM growth, indicating that targeting SATB2/CBP may effectively improve GBM treatment.

**Keywords** CBP; FOXM1; glioblastoma; glioma stem cell; SATB2
**Subject Categories** Cancer; Chromatin, Transcription & Genomics; Neuroscience

## Introduction

Glioblastoma (GBM; WHO grade IV glioma) is the most frequent and malignant type of human primary brain tumor. The prognosis of GBM is extremely poor despite significant advances in the treatment of other solid cancers. The median survival of GBM patients remains less than 16 months (Furnari *et al*, 2007; Stupp *et al*, 2009). The standard therapies including surgical resection, radiation therapy, and chemotherapy are largely ineffective for GBMs due to universal therapeutic resistance and rapid tumor recurrence (Wen & Kesari, 2008). Therefore, it is crucial to identify new treatments to improve the anti-cancer efficacy. GBM displays striking cellular heterogeneity and hierarchy within a tumor containing a fraction of stem cell-like cancer cells called glioma stem cells (GSCs) at the apex of differentiation hierarchy. GSCs exhibit remarkable capacities of proliferation and self-renewal and play critical roles in modulating the tumor microenvironment, neovascularization, cancer invasion, and immune evasion (Magee *et al*, 2012; Lathia *et al*, 2015; Finocchiaro, 2017; Tao *et al*, 2020). Accumulating evidence supports that GSCs are responsible for tumor initiation, progression, and therapeutic resistance (Bao *et al*, 2006; Jin *et al*, 2017). Thus, better understanding of the molecular mechanisms driving GSC proliferation and self-renewal may offer new insights into GBM tumorigenesis, which may lead to effective therapeutic approaches to improve GBM treatment.

Altered chromatin organization is one of hallmarks in cancer cells. Abnormalities in chromatin architecture and transcriptional regulation occur in most cancer cells (He *et al*, 2008), but our knowledge regarding the potential mechanisms driving the changes in chromatin organization and transcription activity in cancer cells

1 Department of Cancer Biology, Lerner Research Institute, Cleveland Clinic, Cleveland, OH, USA
2 Division of Regenerative Medicine, Department of Medicine, University of California, San Diego, San Diego, CA, USA
3 Department of Pathology, Case Western Reserve University School of Medicine, Cleveland, OH, USA
4 Brain Tumor and Neuro-Oncology Center & Center of Excellence for Translational Neuro-Oncology, University Hospitals Seidman Cancer Center, Case Western Reserve University, Cleveland, OH, USA
5 Case Comprehensive Cancer Center, Case Western Reserve University School of Medicine, Cleveland, OH, USA
6 Brain Tumor and Neuro-Oncology Center, Taussig Cancer Institute, Cleveland Clinic, Cleveland, OH, USA
7 Department of Cardiovascular and Metabolic Sciences, Cleveland Clinic, Cleveland, OH, USA
8 Center for Cancer Stem Cell Research, Lerner Research Institute, Cleveland Clinic, Cleveland, OH, USA
9 Department of Radiation Oncology, Taussig Cancer Institute, Cleveland Clinic, Cleveland, OH, USA
*Corresponding author. Tel: +1 216 636 1009; Fax: +1 216 636 5454; E-mail: baos@ccf.org

is limited. The alteration of chromatin organization in cancer cells leads to dysregulation of gene expression, which contributes to the malignant transformation of cells (Schuster-Bockler & Lehner, 2012). Nuclear matrix-associated proteins (NMPs) are a family of proteins that specifically bind to the matrix attachment regions (MAR) of genomic DNA to regulate chromatin organization and gene expression (Dunn *et al*, 2003; Wang *et al*, 2010; Yamaguchi & Takanashi, 2016). Aberrant expression of NMPs has been shown in various human cancers, including breast cancer, lymphoma, colon cancer, non-small cell lung cancer, gastric cancer, and liver cancer (Lever & Sheer, 2010). However, whether NMPs are aberrantly expressed in glioma cells particularly GSCs is not clear, and whether the altered NMP expression contributes to GBM malignant growth has not been defined.

To interrogate the potential relationship between the expression of NMPs and GBM tumor development, we queried the expression pattern of NMPs in clinical database and found that SATB2 (the Special AT-rich Binding Protein-2) is enriched in GBMs. SATB2 is a transcription factor that was originally identified as a protein interacting with the nuclear matrix attachment regions (MAR) of DNA. STAB2 regulates gene expression by modulating chromatin architecture and functioning as a transcriptional cofactor (Dobreva *et al*, 2003; Britanova *et al*, 2008; Diaz-Alonso *et al*, 2012). When SATB2 is localized to the matrix attachment region, it promotes chromatin rearrangement by recruiting chromatin-remodeling proteins to these DNA sequences to either activate or repress gene transcription (Britanova *et al*, 2008; Gyorgy *et al*, 2008; Zhou *et al*, 2012). SATB2 is an evolutionarily conserved protein in vertebrates from zebrafish to mammals (Sheehan-Rooney *et al*, 2010). SATB2 as one of critical NMPs has multiple roles in osteoblast differentiation, craniofacial patterning, cleft palate formation, and neuronal development (Dobreva *et al*, 2006; Leoyklang *et al*, 2007; Britanova *et al*, 2008; Zarate & Fish, 2017). SATB2 affects craniofacial morphogenesis via repression of HOXA2 and regulates osteoblast differentiation by interacting with transcription factors RUNX2 and ATF4 to enhance their activity (Dobreva *et al*, 2006). Moreover, SATB2 regulates neuronal specification during development by recruiting the chromatin-remodeling complexes to *CTIP2* locus to regulate its expression (Britanova *et al*, 2008). Recent studies demonstrated that SATB2 is associated with tumor growth or suppression. SATB2 has been shown to suppress tumor progression in colorectal cancer, non-small cell lung cancer and gastric cancer, and high SATB2 expression is associated with a favorable prognosis (Mansour *et al*, 2015; Wu *et al*, 2016; Ma *et al*, 2018). However, SATB2 promotes tumor growth in hepatocellular carcinoma, osteosarcoma, and triple-negative breast cancer (Jiang *et al*, 2015; Luo *et al*, 2016; Xu *et al*, 2017). Our analysis of clinical database indicated that SATB2 is enriched in GBMs, suggesting that SATB2 may play a tumor-promoting role in GBMs. Therefore, we investigated the potential role of SATB2 in regulating GSC properties and GBM tumor growth, and found that SATB2 augmented GSC proliferation by recruiting histone acetyltransferase CBP to promote FOXM1 expression in GSCs.

FOXM1 is a member of the forkhead box transcription factor family, which is evolutionarily conserved and contains a common DNA-binding domain called forkhead box domain (Laoukili *et al*, 2007; Liao *et al*, 2018). FOXM1 is a typical transcription factor related to cell proliferation and involved in cancer growth (Laoukili

*et al*, 2007; Nandi *et al*, 2018). It regulates cell cycle process by modulating the expression of various cell cycle-related genes required for G1/S, G2/M progression (Li *et al*, 2012). FOXM1 is highly expressed in GBM and informs poor prognosis of GBM patients (Liu *et al*, 2006; Lee *et al*, 2015; Zhong *et al*, 2016). However, the functional significance and the molecular mechanisms underlying FOXM1 regulation in GSCs are poorly understood. In this study, we found that FOXM1 expression is activated by SATB2 in GSCs. SATB2 binds to the MAR sequence of the *FOXM1* gene locus and recruiting CBP to the MAR site to promote FOXM1 expression. Our study uncovers a critical role of the SATB2/CBP complex in regulating FOXM1 expression to promote GSC proliferation and GBM malignant growth. Importantly, inhibition of SATB2/CBP transcriptional activity by the CBP inhibitor C646 significantly suppressed GSC proliferation and GBM tumor growth, indicating that targeting SATB2/CBP may be an effective therapeutic strategy to improve GBM treatment.

## Results

### SATB2 is preferentially expressed by GSCs

To determine the potential relationship between the nuclear matrix-associated proteins (NMPs) and GBM malignant growth, we mapped the expression of NMPs in TCGA GBM and low-grade glioma (LGG) databases, with consideration of tumor transcriptional subtype, *IDH1/P53/PTEN* mutation status, tumor grade, patient age, and performance status. Our analyses focused on several key NMPs including SATB1/2, SAFB1/2, EZH2, SUZ12, BMI1, PCL3, RAE28, and CTCF, as these NMPs have been shown to be aberrantly expressed in cancers (Lever & Sheer, 2010). The analyses revealed that SATB2, EZH2, SUZ12, and PCL3 are enriched in older patients with glioblastoma (GBM) with worse performance status (Appendix Fig S1). Among these four genes, SATB2's role in GBM progression has not been defined. To interrogate the functional significance of SATB2 expression in GBM malignant growth, we initially examined SATB2 expression pattern in several human GBM specimens and found that SATB2 is preferentially expressed in nuclei of glioma cells expressing the GSC markers SOX2 and OLIG2 (Fig 1A and B; Appendix Fig S2A). Further experiments demonstrated that SATB2 is rarely expressed in glioma cells expressing the differentiation markers (GFAP, TUBB3, and GALC) in human GBMs (Appendix Fig S2B–G). To confirm the preferential expression of SATB2 in GSCs, we assessed SATB2 expression in isolated GSCs and matched non-stem tumor cells (NSTCs) that were functionally validated as described in Materials and Methods. The results showed that SATB2 and the GSC markers SOX2 and OLIG2 were preferentially expressed in all isolated GSC populations relative to matched NSTCs (Fig 1C–E; Appendix Fig S2H). In addition, SATB2 was expressed at much higher levels in GSCs than in neural progenitor cells (NPCs) (Fig 1F and G). As GSC population decreases during differentiation, we examined the expression of SATB2 during GSC differentiation induced by the serum. A gradual reduction of SATB2 and the GSC marker SOX2 was observed during GSC differentiation, which was accompanied by the increased expression of the differentiation marker GFAP (Fig 1H), indicating a potential link between SATB2 expression and GSC status. Collectively, these data

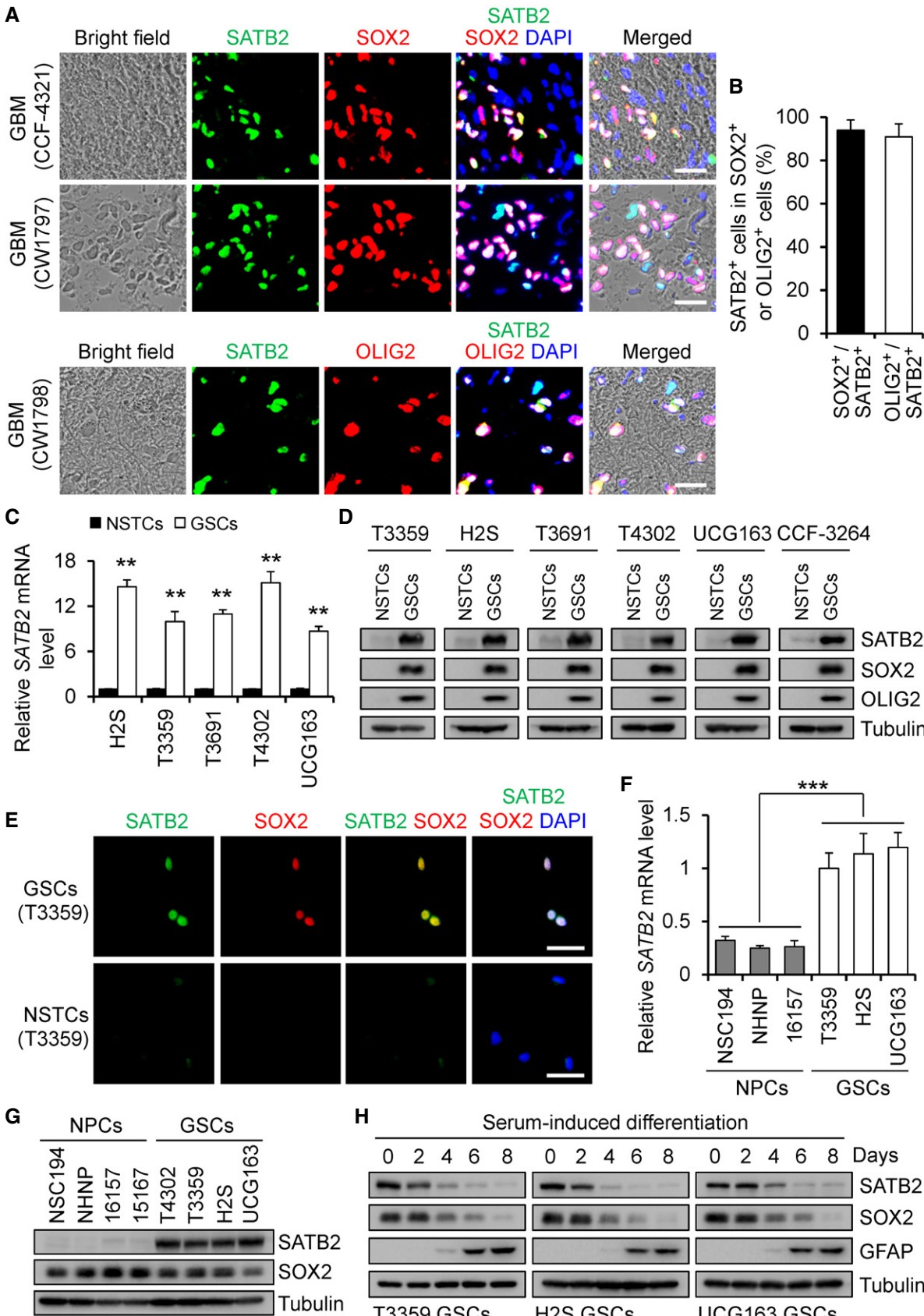

Figure 1.

**Figure 1. SATB2 is preferentially expressed by GSCs.**

A   Immunofluorescence of SATB2 (green) and the GSC marker SOX2 or OLIG2 (red) on frozen tissue sections of human GBM surgical specimens. SATB2 is preferentially expressed by GSCs in human GBMs. Scale bar, 25 μm.

B   Quantification of the fraction of SATB2$^+$ cells in SOX2$^+$ or OLIG2$^+$ cells in human GBMs. More than 90% SOX2$^+$ or OLIG2$^+$ cells showed SATB2 staining. $n = 3$ GBMs.

C   qPCR analysis of *SATB2* mRNA expression in GSCs and matched non-stem tumor cells (NSTCs) ($n = 5$).

D   Immunoblot analysis of SATB2, SOX2, and OLIG2 expression in cell lysates of GSCs and matched NSTCs.

E   Immunofluorescence of SATB2 (green) and SOX2 (red) in T3359 GSCs and matched NSTCs. Scale bar, 50 μm.

F   qPCR analysis of *SATB2* mRNA expression in GSCs and neural progenitor cells (NPCs) ($n = 3$).

G   Immunoblot analysis of SATB2 and SOX2 expression in cell lysates of GSCs and NPCs.

H   Immunoblot analysis of SATB2, GSC marker SOX2, and differentiation marker GFAP expression during serum-induced GSC differentiation.

Data information: Data are represented as mean ± SD. **$P < 0.01$, ***$P < 0.001$, Mann–Whitney test (C) or one-way ANOVA analysis followed by Tukey's test (F). Exact $P$ values are reported in Appendix Table S3.

demonstrate that SATB2 is preferentially expressed by GSCs in GBMs, suggesting a potential role of SATB2 in the GSC maintenance.

## SATB2 is required for GSC proliferation and self-renewal

As SATB2 is preferentially expressed in GSCs, we next investigated the functional significance of SATB2 in the GSC maintenance by using two distinct shRNAs targeting SATB2. Lentivirus-mediated expression of shSATB2-1 or shSATB2-2 markedly reduced SATB2 protein levels in GSCs (Fig 2A). We found that disruption of SATB2 significantly inhibited GSC growth as measured by cell titer assay (Fig 2B) and reduced DNA replication as assayed by EDU incorporation assay (Fig 2C and D). In contrast, disruption of SATB2 had little effect on the growth and survival of NSTCs (Appendix Fig S3A) and NPCs (Appendix Fig S3B). In addition, silencing SATB2 impaired GSC self-renewal as assessed by tumorsphere formation assays (Fig 2E–G) and *in vitro* limiting dilution assays (Fig 2H). Consistently, disrupting SATB2 also reduced expression of the GSC markers including SOX2 and OLIG2 (Appendix Fig S3C). Taken together, these results indicate that SATB2 is essential for GSC proliferation and self-renewal.

## Silencing SATB2 suppresses GSC-driven tumor growth

As the most important property of GSCs is their potent capacity to propagate tumors *in vivo*, we then examined the impact of SATB2 inhibition on the GSC-driven intracranial tumor growth. GSCs (T3359 or H2S) expressing firefly luciferase and shSATB2 (shSATB2-1 or shSATB2-2) or shNT were transplanted into the brains of immunocompromised mice by intracranial injection. In

*vivo* bioluminescent imaging of orthotopic tumors showed that silencing SATB2 dramatically inhibited GSC-driven tumor growth in mouse brains (Fig 3A and B; Appendix Fig S3D and E). Consequently, animals bearing xenografts derived from GSCs expressing shSATB2 survived significantly longer than the control animals (Fig 3C; Appendix Fig S3F). Further experiment demonstrated that silencing SATB2 reduced the tumorogenic potential of GSCs in an *in vivo* limiting dilution assay (Appendix Table S1). Given that SATB2 promotes GSC proliferation *in vitro*, we next examined the impact of SATB2 disruption on cell proliferation in GSC-derived xenografts. Immunofluorescent staining confirmed that the expression of SATB2 was significantly decreased in xenografts expressing shSATB2 (Fig 3D and E; Appendix Fig S3G and H). We found that cell proliferation was markedly reduced in the GBM xenografts derived from GSCs expressing shSATB2 than control tumors as demonstrated by Ki67 immunofluorescence (Fig 3F and G; Appendix Fig S3I and J). In addition, silencing SATB2 significantly reduced GSC population as revealed by SOX2 immunofluorescence (Fig 3H and I; Appendix Fig S3K and L). Collectively, these data demonstrate that SATB2 disruption potently suppresses the GSC-driven tumor growth, indicating that SATB2 is critical for maintaining the tumorigenic potential of GSCs *in vivo*.

## SATB2 is required for the expression of genes involved in cell cycle progression

To understand the molecular mechanisms by which SATB2 promotes GSC proliferation, we performed transcriptional profiling on GSCs expressing shSATB2 or shNT and found that GSCs expressing shSATB2 exhibited distinct gene expression profiles relative to control GSCs with shNT (Fig 4A). Gene ontology analysis suggested

**Figure 2. Disrupting SATB2 impaired GSC proliferation and self-renewal.**

A   Immunoblot analysis of SATB2 expression in GSCs transduced with lentiviral-mediated non-targeting shRNA (shNT) or SATB2 shRNA (shSATB2).

B   Cell viability of GSCs transduced with shNT or shSATB2 ($n = 5$).

C   EdU incorporation assay of GSCs transduced with shNT or shSATB2. Scale bar: 50 μm.

D   Quantification of (C) showing the percentage of EdU$^+$ cells ($n = 5$).

E   Tumorsphere images of GSCs transduced with shNT or shSATB2. Scale bar: 100 μm.

F, G   Quantification of the diameter (F) or number (G) of tumorspheres formed by GSCs expressing shNT or shSATB2 (F: $n = 9$; G: $n = 5$).

H   *In vitro* limiting dilution analysis of the tumorsphere formations of GSCs expressing shNT or shSATB2. Silencing SATB2 attenuated the self-renewal capacity of GSCs.

Data information: Data are represented as mean ± SD. ****$P < 0.0001$, two-way ANOVA analysis followed by Tukey's test (B), one-way ANOVA analysis followed by Tukey's test (D, F, and G) or ELDA analysis for differences in stem cell frequencies (H). Exact $P$ values are reported in Appendix Table S3.

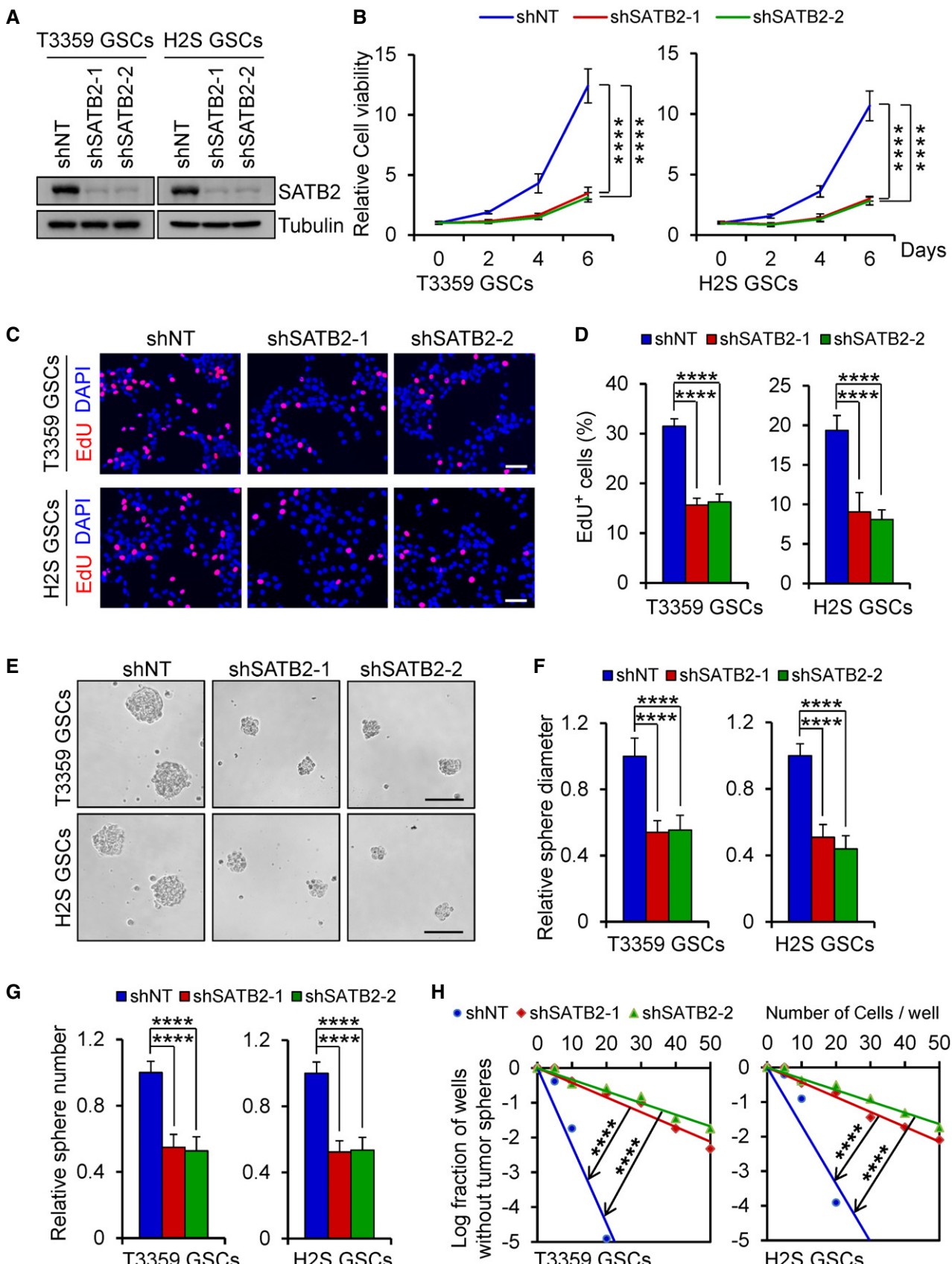

Figure 2.

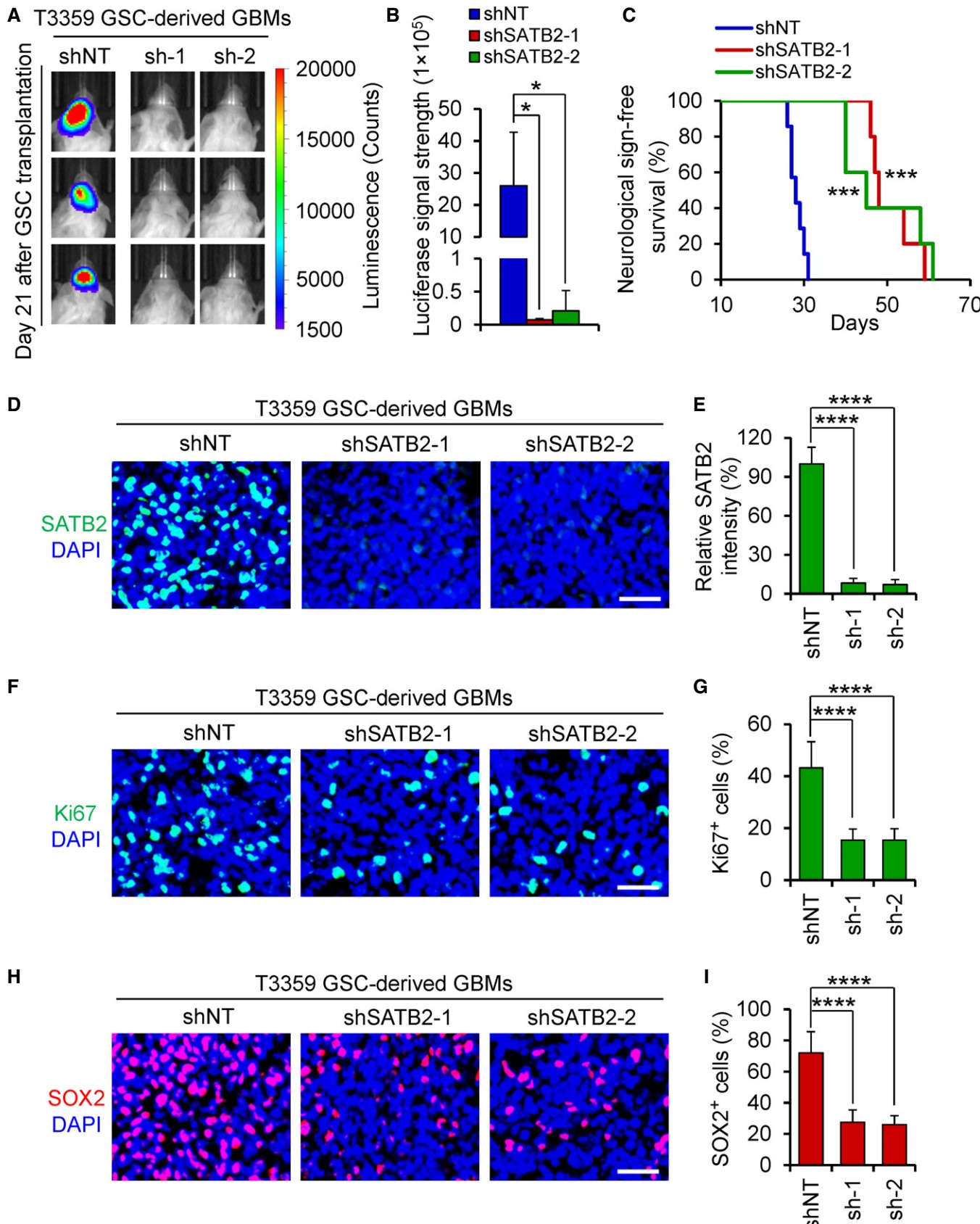

Figure 3.

**Figure 3.  Disrupting SATB2 inhibited tumor growth and prolonged mice survival.**

A   Bioluminescent images of the GBM xenografts derived from the luciferase-labeled T3359 GSCs expressing NT or SATB2 shRNA. Representative images on day 21 posttransplantation are shown (*n* = 5 mice per group). Silencing SATB2 significantly delayed GBM growth.

B   Quantification of the bioluminescence of xenografts derived from the luciferase-labeled T3359 GSCs expressing shNT or shSATB2 on day 21 posttransplantation (*n* = 5 mice per group).

C   Kaplan–Meier survival curves of mice intracranially implanted with T3359 GSCs expressing shNT or shSATB2 (shNT: *n* = 7 mice; shSATB2-1 or shSATB2-2: *n* = 5 mice). Median survival: shNT, 28 days; shSATB2-1, 48 days; shSATB2-2, 45 days. Animals bearing GSC-derived xenografts expressing SATB2 shRNA survived longer than the control animals.

D   Immunofluorescence of SATB2 (Green) in xenografts derived from T3359 GSCs expressing shNT or shSATB2 (*n* = 5 tumors per group). Scale bar: 40 μm.

E   Quantification of SATB2 intensity in xenografts derived from T3359 GSCs expressing shNT or shSATB2 (*n* = 5 tumors per group).

F   Immunofluorescence of Ki67 (Green) in tumor xenografts derived from T3359 GSCs expressing shNT or shSATB2 (*n* = 5 tumors per group). Scale bar: 40 μm.

G   Quantification of Ki67 positive cells in xenografts derived from T3359 GSCs expressing shNT or shSATB2 (*n* = 5 tumors per group).

H   Immunofluorescence of SOX2 (Red) in xenografts derived from T3359 GSCs expressing shNT or shSATB2 (*n* = 4 tumors per group). Scale bar: 40 μm.

I   Quantification of SOX2 positive cells in xenografts derived from T3359 GSCs expressing shNT or shSATB2 (*n* = 4 tumors per group).

Data information: Data are shown as mean ± SD. *$P < 0.05$, ***$P < 0.001$, ****$P < 0.0001$ compared with shNT group, Kruskal–Wallis test followed by Dunn's test (B), log-rank test (C), or one-way ANOVA analysis followed by Tukey's test (E, G, and I). Exact *P* values are reported in Appendix Table S3.

that the most significantly down-regulated expression of genes in GSCs after SATB2 disruption is associated with cell cycle progression and chromosome organization (Fig 4B), while there was no significant functional relevance for the up-regulated genes in GSCs expressing shSATB2. Among the genes involved in cell cycle process, FOXM1 appeared to be the most important downstream target of SATB2 due to several reasons. First, FOXM1 is a master transcription factor in cell cycle regulation and proliferation (Wierstra & Alves, 2007; Li *et al*, 2012). Second, FOXM1 is overexpressed in GBMs and informs poor survival of GBM patients (Lee *et al*, 2015; Zhong *et al*, 2016). Lastly, FOXM1 is co-localized with GSC markers SOX2 and Nestin in primary GBM specimens (Joshi *et al*, 2013). Therefore, we validated the effect of SATB2 disruption on *FOXM1* expression using qPCR analysis. As expected, the mRNA expression of *FOXM1* was markedly reduced by SATB2 disruption in GSCs (Fig 4C). Immunoblot analysis confirmed that knockdown of SATB2 resulted in a reduction in FOXM1 protein levels in GSCs (Fig 4D). In addition, decreased FOXM1 expression was detected in GSC-derived xenografts expressing shSATB2 relative to the shNT control (Fig 4E). Consistently, expression levels of FOXM1 transcriptional targets were also reduced by SATB2 disruption in GSCs as indicated by the gene expression profiling analyses (Fig 4F). qPCR analysis further confirmed that knockdown of SATB2 decreased expression of these genes in GSCs (Fig 4G), while disruption of SATB2 increased the expression of *p21* and *p27* which are negatively regulated by FOXM1 (Fig 4G). These data suggest that

SATB2 may mediate through FOXM1 to regulate GSC proliferation, self-renewal, and tumorigenic potential.

## SATB2 promotes GSC proliferation and tumor propagation through FOXM1

As FOXM1 is an oncogenic regulator that promotes GSC proliferation and expression of the stem cell marker SOX2 (Lee *et al*, 2015), and SATB2 regulates FOXM1 expression, we next explored whether FOXM1 mediates the effects of SATB2 on GSC proliferation and tumor growth. To address this possibility, we examined whether ectopic expression of FOXM1 could rescue the effects impaired by SATB2 disruption. We established a lentiviral vector expressing FOXM1 and introduced it into GSCs expressing shSATB2 or shNT control (Fig 5A; Appendix Fig S4A). Ectopic expression of FOXM1 largely rescued the impaired proliferation and tumorsphere formation of GSCs caused by SATB2 disruption (Fig 5B and C; Appendix Fig S4B and C). In addition, forced expression of FOXM1 promoted the re-expression of proliferation-related genes (Appendix Fig S4D). Consistently, ectopic expression of FOXM1 in GSCs expressing shSATB2 restored GBM tumor growth and attenuated the increased survival of mice bearing the GSC-derived GBMs (Fig 5D–F). Further analysis showed that forced expression of FOXM1 rescued the impaired *in vivo* cell proliferation of GSCs expressing shSATB2 as marked by Ki67 immunofluorescence (Fig 5G and H). Furthermore, ectopic expression of FOXM1 restored

**Figure 4.  SATB2 regulates cell cycle gene expression.**

A   Heatmap analysis of differentially expressed genes between SATB2 silencing H2S GSCs (shSATB2) and control H2S GSCs (shNT). Differentially expressed genes had a 1.8-fold or greater expression difference. Among differentially expressed genes, 160 are upregulated and 185 are downregulated.

B   Gene ontology analysis of genes downregulated in SATB2 silencing GSCs compared with control GSCs. The most significantly down-regulated genes are associated with cell cycle progression and chromosome organization. *P* values were generated using the PANTHER tool (http://pantherdb.org/).

C   qPCR analysis of *FOXM1* and *SATB2* mRNA expression in GSCs transduced with shNT or shSATB2 (*n* = 3).

D   Immunoblot analysis of FOXM1 and SATB2 expression in GSCs transduced with shNT or shSATB2.

E   Immunoblot analysis of FOXM1 and SATB2 expression in tumor xenografts derived from T3359 GSCs expressing shNT or shSATB2.

F   Heatmap analysis of FOXM1 downstream targets between SATB2 silencing H2S GSCs (shSATB2) and control H2S GSCs (shNT) from microarray analysis. Differentially expressed genes had a 1.5-fold or greater expression difference.

G   qPCR analysis of FOXM1 downstream targets in H2S GSCs transduced with shNT or shSATB2 (*n* = 3).

Data information: Data are represented as mean ± SD. *$P < 0.05$, **$P < 0.01$, ***$P < 0.001$, ****$P < 0.0001$, one-way ANOVA analysis followed by Tukey's test. Exact *P* values are reported in Appendix Table S3.

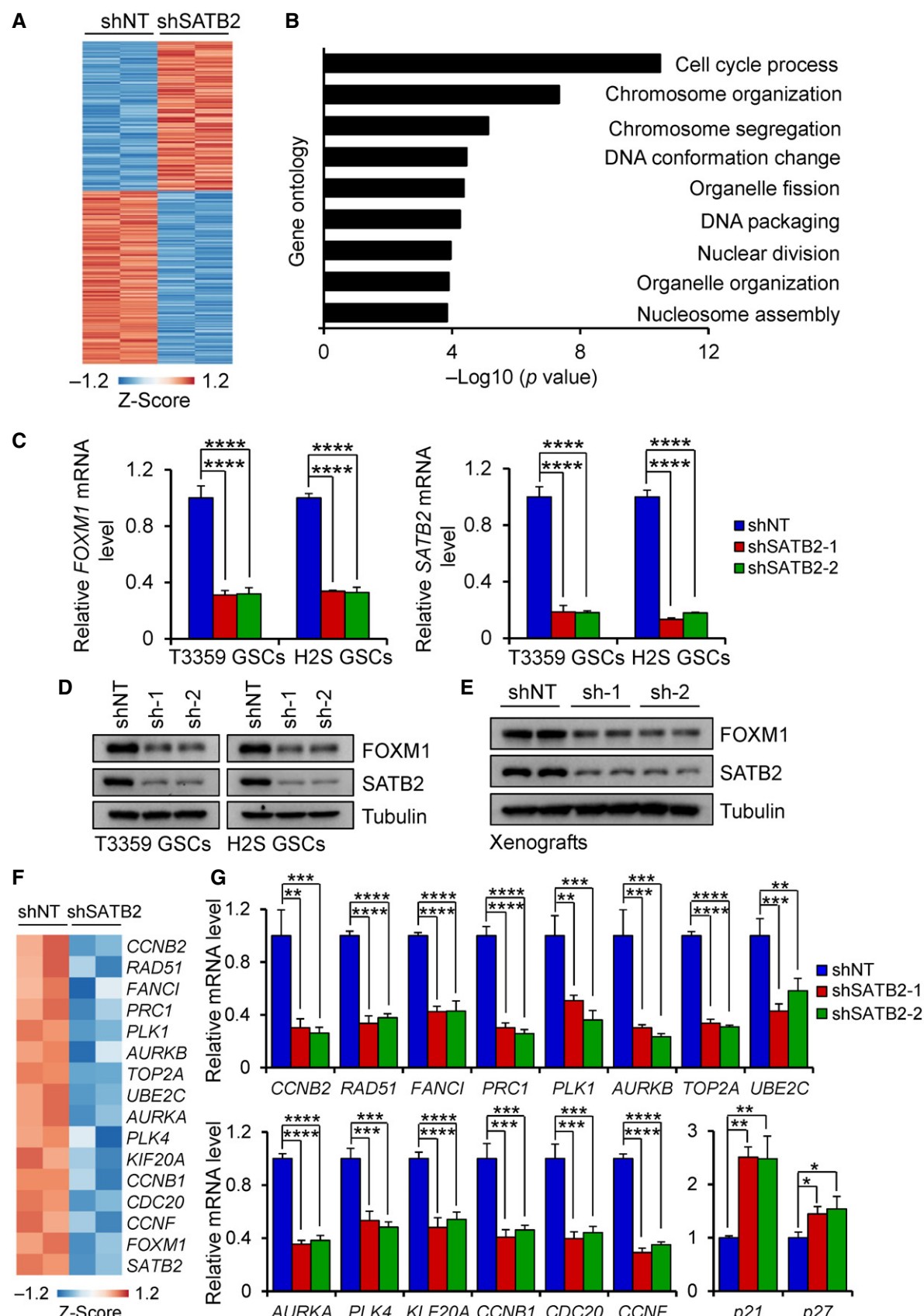

**Figure 4.**

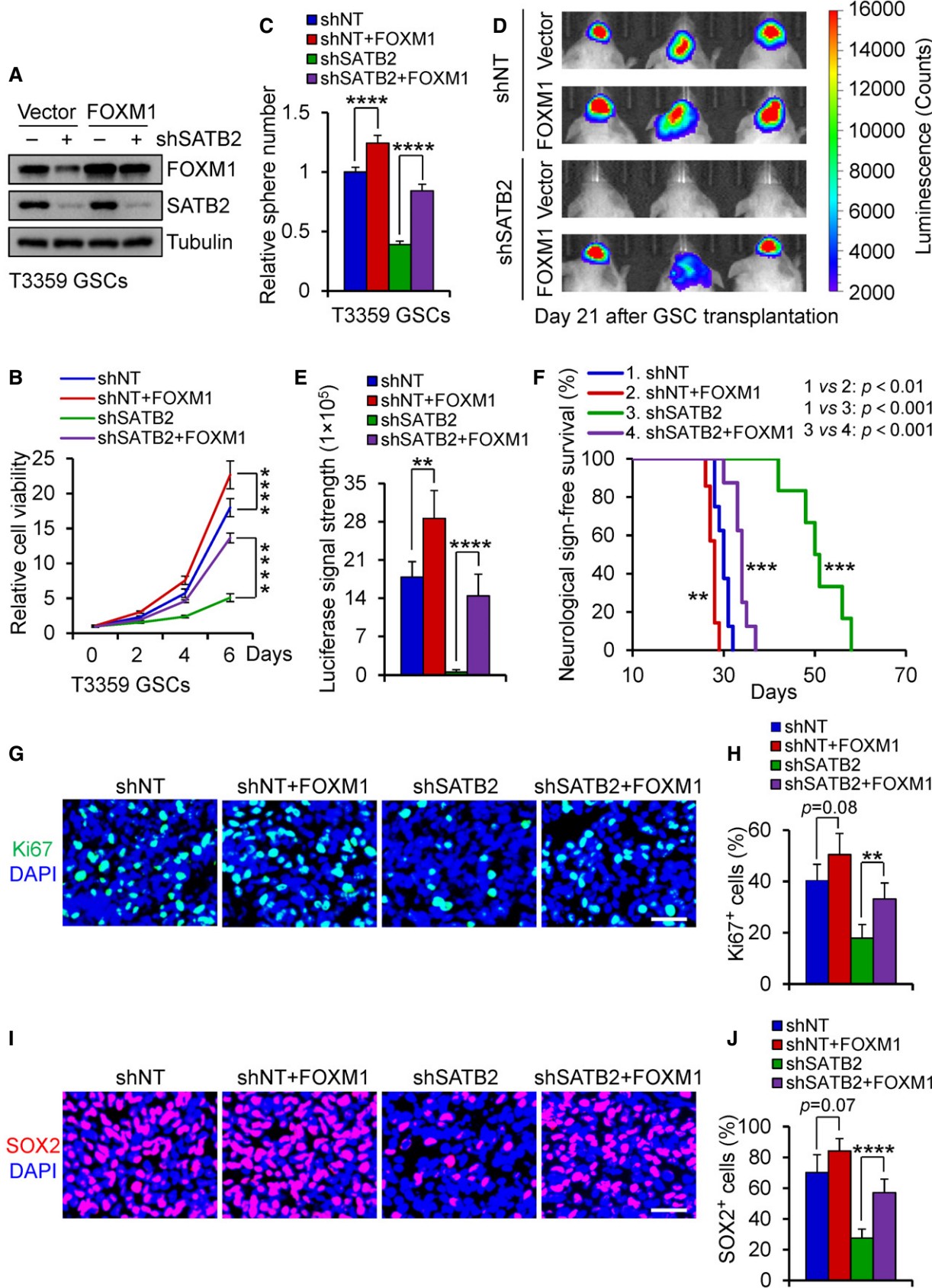

Figure 5.

**Figure 5. Ectopic expression of FOXM1 rescued the decreased GSC proliferation and GBM growth caused by SATB2 disruption.**

A   Immunoblot analysis of FOXM1 and SATB2 expression in T3359 GSCs transduced with FOXM1 or vector control in combination with shNT or shSATB2.
B   Cell viability assay of T3359 GSCs transduced with FOXM1 or vector control in combination with shNT or shSATB2 ($n = 5$). Ectopic expression of FOXM1 restored the cell proliferation impaired by SATB2 silencing.
C   Tumorsphere number of T3359 GSCs transduced with FOXM1 or vector control in combination with shNT or shSATB2 ($n = 5$). Ectopic expression of FOXM1 restored the tumorsphere formation of GSCs impaired by SATB2 silencing.
D, E  *In vivo* bioluminescent images (D) or quantification (E) of the tumor xenografts derived from luciferase-labeled T3359 GSCs transduced with FOXM1 or vector control in combination with shNT or shSATB2 (shNT: $n = 5$ mice; shNT + FOXM1: $n = 4$ mice; shSATB2: $n = 5$ mice; shSATB2 + FOXM1: $n = 5$ mice). Representative images on day 21 posttransplantation are shown. Ectopic expression of FOXM1 in GSCs expressing shSATB2 markedly restored GBM tumor growth.
F   Kaplan–Meier survival curves of mice intracranially implanted with T3359 GSCs transduced with FOXM1 or vector control in combination with shNT or shSATB2 (shNT: $n = 8$ mice; shNT + FOXM1: $n = 7$ mice; shSATB2: $n = 6$ mice; shSATB2 + FOXM1: $n = 8$ mice). Median survival: shNT, 30 days; shNT + FOXM1, 28 days; shSATB2, 50.5 days; shSATB2 + FOXM1, 34 days. Ectopic expression of FOXM1 in GSCs expressing shSATB2 markedly attenuated the increased survival of mice bearing the GSC-derived GBMs.
G   Immunofluorescence of Ki67 (Green) in xenografts derived from T3359 GSCs transduced with FOXM1 or vector control in combination with shNT or shSATB2 (shNT: $n = 6$ tumors; shNT + FOXM1: $n = 5$ tumors; shSATB2: $n = 6$ tumors; shSATB2 + FOXM1: $n = 6$ tumors). Scale bar: 40 μm.
H   Quantification of Ki67 positive cells in xenografts derived from T3359 GSCs transduced with FOXM1 or vector control in combination with shNT or shSATB2 (shNT: $n = 6$ tumors; shNT + FOXM1: $n = 5$ tumors; shSATB2: $n = 6$ tumors; shSATB2 + FOXM1: $n = 6$ tumors).
I   Immunofluorescence of SOX2 (Red) in xenografts derived from T3359 GSCs transduced with FOXM1 or vector control in combination with shNT or shSATB2 (shNT: $n = 6$ tumors; shNT + FOXM1: $n = 5$ tumors; shSATB2: $n = 6$ tumors; shSATB2 + FOXM1: $n = 6$ tumors). Scale bar: 40 μm.
J   Quantification of SOX2 positive cells in xenografts derived from T3359 GSCs transduced with FOXM1 or vector control in combination with shNT or shSATB2 (shNT: $n = 6$ tumors; shNT + FOXM1: $n = 5$ tumors; shSATB2: $n = 6$ tumors; shSATB2 + FOXM1: $n = 6$ tumors).

Data information: Data are shown as mean ± SD. **$P < 0.01$, ***$P < 0.001$, ****$P < 0.0001$, two-way ANOVA analysis followed by Tukey's test (B), one-way ANOVA analysis followed by Tukey's test (C, E, H, and J) or log-rank test (F). Exact $P$ values are reported in Appendix Table S3.

GSC population in xenografts expressing shSATB2 (Fig 5I and J). Immunofluorescent staining confirmed a significant reduction of SATB2 expression in xenografts expressing shSATB2 (Appendix Fig S4E and F). These results demonstrate that SATB2 activates FOXM1 to promote GSC proliferation and the GSC-driven GBM tumor growth.

## SATB2 binds to the MAR sequence of FOXM1 gene locus and recruits the coactivator CBP

We next explored how SATB2 regulates the expression of FOXM1. SATB2 has been shown to regulate gene expression by binding to the matrix attachment region (MAR) of target genes (Dobreva *et al*, 2003; Diaz-Alonso *et al*, 2012). To investigate whether FOXM1 is directly regulated by SATB2, we analyzed a ~ 20 kb region upstream and downstream of the gene's first exon to identify potential SATB2 binding by using the Marscan tool. This tool searches for a MAR recognition signature that is a specific 8bp sequence

(AATAAYAA) and a 16 bp sequence (AWWRTAANNWWGNNNC) within a 200 bp distance from each other. The analysis identified a single MAR recognition signature, consists of the typical 8bp sequence (AATAACAA) located 19.839–19.846 kb upstream of the first exon of *FOXM1* gene and the 16 bp sequence (ATTTTAA-CAATGTTTC) located 19.823–19.838 kb upstream of the first exon, suggesting that there is a potential MAR region on the *FOXM1* gene locus. To assess whether SATB2 can bind to this specific MAR, we used chromatin immunoprecipitation (ChIP) assays and found that SATB2 bound to this MAR sequence on the *FOXM1* gene in GSCs, while the binding is extremely weak in NSTCs or NPCs (Fig 6A and B). In addition, silencing SATB2 attenuated this binding in GSCs (Fig 6C). These data indicate that SATB2 may directly regulate *FOXM1* expression by binding to the MAR region. Previous reports have shown that SATB2 activates expression of downstream genes by recruiting histone acetyltransferase P300 (Rainger *et al*, 2014; Wang *et al*, 2019). To determine whether SATB2 recruits P300 to regulate FOXM1 expression, we first performed co-

**Figure 6. SATB2 binds to the MAR sequence in the *FOXM1* gene locus and recruits CBP acetyltransferase in GSCs.**

A   Schematic representation of MAR (Matrix Attachment Region) within *FOXM1* locus. Arrows show the location of PCR primers for ChIP experiments.
B   ChIP assays with the SATB2 antibody or IgG using GSCs, NSTCs, and NPCs. PCR primers amplified a fragment flanking the MAR of *FOXM1* gene locus. Note that abundant SATB2 binds to the MAR of *FOXM1* gene locus in GSCs.
C   qPCR analysis of ChIP assays with the SATB2 antibody or IgG using GSCs transduced with shNT or shSATB2 ($n = 3$). PCR primers amplified a fragment flanking the MAR of *FOXM1* gene locus. Silencing SATB2 decreased its binding amount to the MAR of *FOXM1* gene locus.
D   CoIP assays of endogenous protein interaction in GSCs. Immunoblots of precipitated proteins or total lysates were performed using indicated antibodies. Note that SATB2 associates with endogenous CBP while not P300.
E   qPCR analysis of ChIP assays with the CBP antibody or IgG using GSCs transduced with shNT or shSATB2 ($n = 3$). PCR primers amplified a fragment flanking the MAR of *FOXM1* gene locus. Silencing SATB2 reduced the binding of CBP to the MAR of *FOXM1* gene locus.
F   qPCR analysis of ChIP assays with the indicated antibody (AcH3K18, AcH3K27, AcH4) using T3359 GSCs transduced with shNT or shSATB2 ($n = 3$). PCR primers amplified a fragment flanking the MAR of *FOXM1* gene locus. Silencing SATB2 reduced acetylation of H3K18, H3K27, and H4 levels on the MAR of *FOXM1* locus.
G   qPCR analysis of *FOXM1* mRNA expression in T3359 GSCs transduced with shSATB2 or shCBP or both ($n = 3$).
H   Immunoblot analysis of FOXM1 expression in T3359 GSCs transduced with shSATB2 or shCBP or both.

Data information: Data are shown as mean ± SD. *$P < 0.05$, **$P < 0.01$, ***$P < 0.001$, ****$P < 0.0001$, one-way ANOVA analysis followed by Tukey's test. Exact $P$ values are reported in Appendix Table S3.

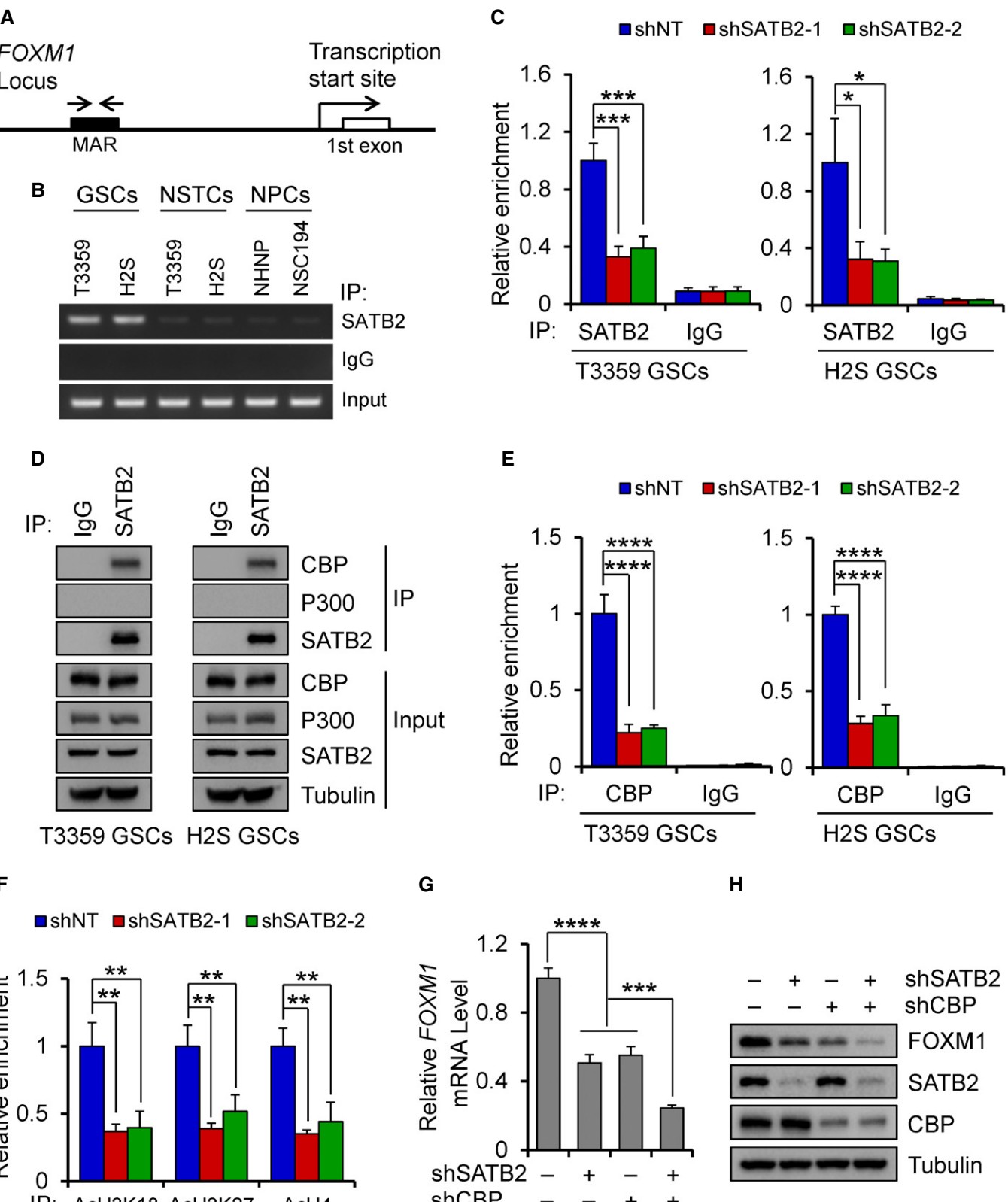

**Figure 6.**

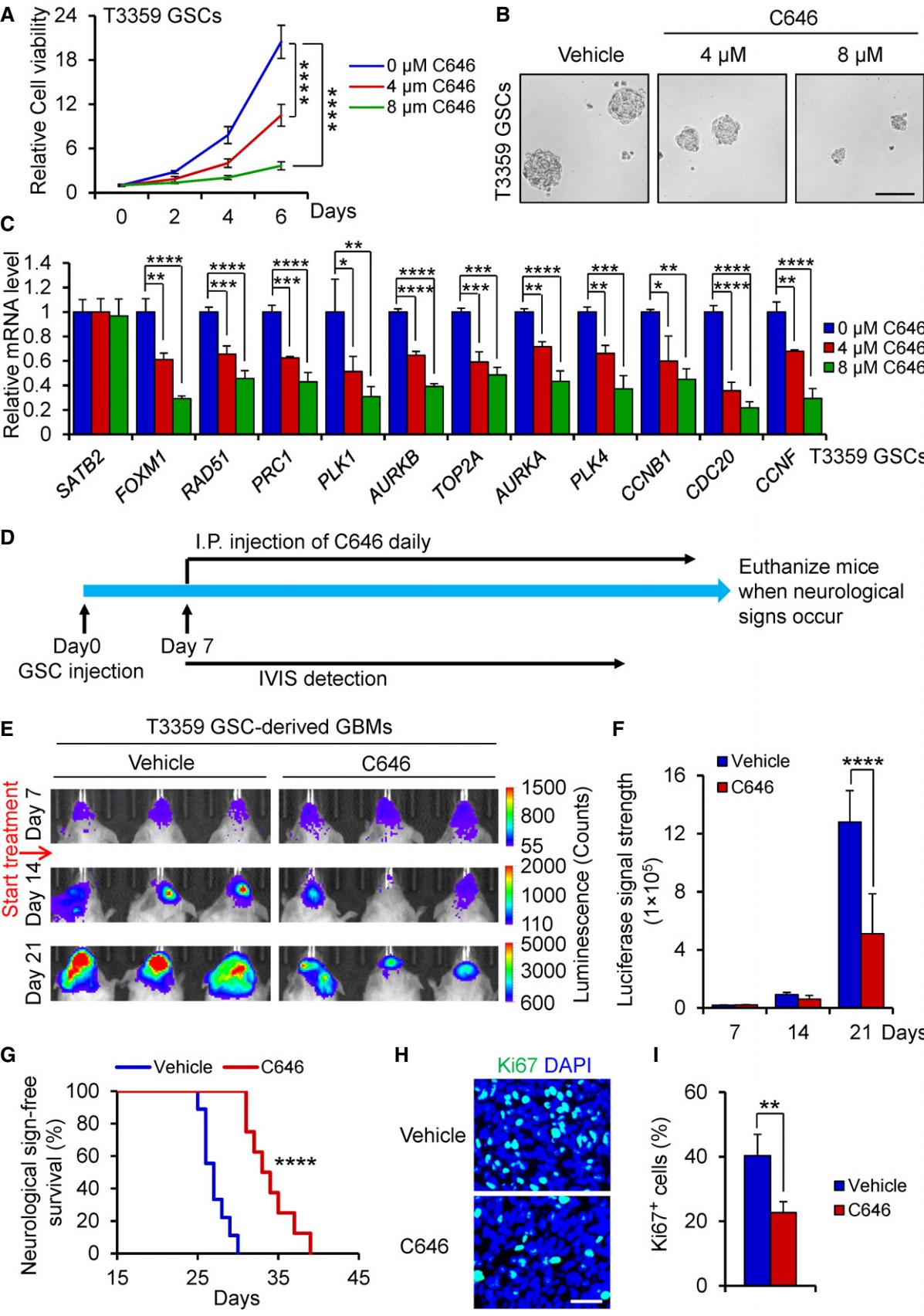

Figure 7.

**Figure 7.  C646 treatment inhibited GSC proliferation and GSC-driven tumor growth.**

A   Cell viability of T3359 GSCs treated with indicated doses of C646 or the vehicle control (*n* = 5).

B   Tumorsphere images of T3359 GSCs treated with indicated doses of C646 or the vehicle control for 6 days. Scale bar: 100 µm.

C   qPCR analysis of *SATB2*, *FOXM1*, and FOXM1 downstream targets in T3359 GSCs treated with indicated doses of C646 or the vehicle control for 24 h (*n* = 3).

D   Schematic diagram showing the treatment of mice bearing the GSC-derived xenografts with C646. After GSC transplantation for 7 days, mice were treated with C646 or the vehicle control daily. Mice were monitored by IVIS bioluminescent imaging and maintained until neurological signs occur.

E   Bioluminescent imaging of tumor growth in mice bearing xenografts derived from the luciferase-labeled T3359 GSCs treated with C646 or the vehicle control at indicated days after GSC transplantation (*n* = 5 mice per group).

F   Quantification of tumor growth from (E) (*n* = 5 mice per group).

G   Kaplan–Meier survival curves of mice bearing T3359 GSC-derived xenografts treated with C646 or the vehicle control (Vehicle control: *n* = 9 mice; C646: *n* = 8 mice). Median survival: Vehicle control, 27 days; C646, 33.5 days.

H   Immunofluorescence of Ki67 (Green) in T3359 GSC-derived xenografts from mice treated with C646 or the vehicle control (*n* = 6 tumors per group). Scale bar: 40 µm.

I   Quantification of Ki67 positive cells in T3359 GSC-derived xenografts from mice treated with C646 or the vehicle control (*n* = 6 tumors per group).

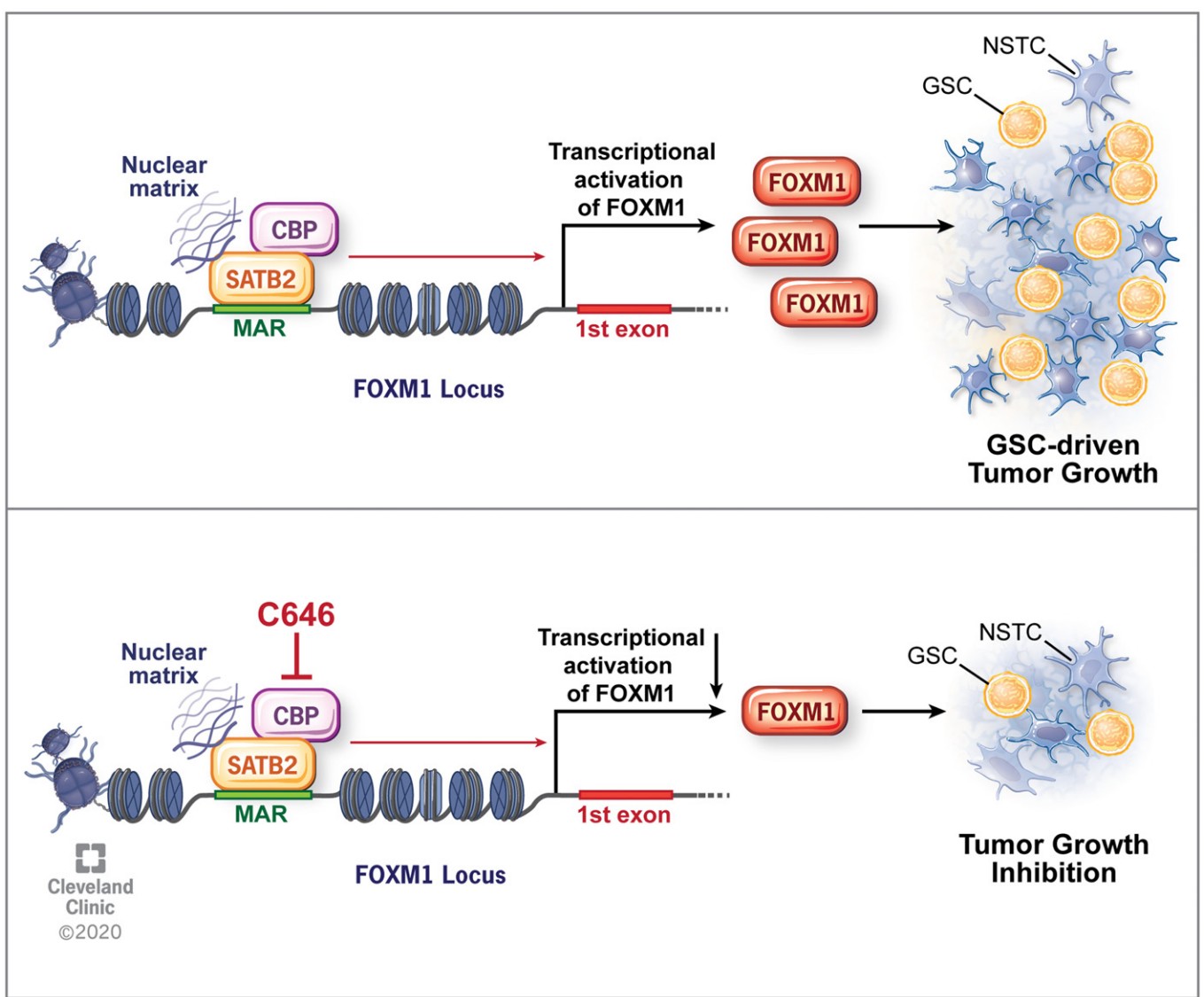

**Figure 8.   A schematic model showing the function of SATB2 and CBP in promoting FOXM1 expression and GSC-driven tumor growth.**

SATB2 binds to the MAR region of its target gene *FOXM1* to remodel local chromatin by recruiting histone acetyltransferase CBP, which in turn promotes *FOXM1* transcription, leading to GSC proliferation and GBM tumor growth. Inhibition of SATB2/CBP transcriptional activity by the CBP inhibitor C646 suppressed GSC proliferation and tumor growth. Reprinted with permission, Cleveland Clinic Center for Medical Art & Photography © 2020. All Rights Reserved.

immunoprecipitation (CoIP) assay to confirm their binding. CoIP experiments showed that SATB2 was not bound to P300 in GSCs, while SATB2 bound to the P300 homolog CBP (Fig 6D). In addition, disrupting SATB2 indeed reduced the binding of CBP to the MAR of the *FOXM1* gene locus (Fig 6E). Hyperacetylation of histones is associated with transcriptional activation (MacDonald & Howe, 2009). Previous studies have shown that CBP preferentially acetylates histone H3K18/K27 and H4 to regulate chromatin function and gene transcription (Ogryzko *et al*, 1996; Jin *et al*, 2011; Lasko *et al*, 2017). As SATB2 recruits CBP to the MAR of *FOXM1* locus, we next asked whether chromatin state at the MAR was affected by SATB2 disruption. ChIP-qPCR analyses demonstrated that silencing SATB2 indeed markedly reduced the acetylation of H3K18, H3K27, and H4 levels on the MAR of *FOXM1* locus (Fig 6F), indicating that the SATB2/CBP complex could alter local chromatin structure from a repressive to an active state. Because SATB2 binds to CBP to regulate FOXM1 expression, we next explore whether SATB2 and CBP have a synergistic effect on regulating FOXM1 expression. qPCR and immunoblot analysis showed that double knockdown of SATB2 and CBP further decreased FOXM1 mRNA and protein levels relative to single knockdown in GSCs (Fig 6G and H). Taken together, these data suggest that SATB2 activates FOXM1 expression by binding to MAR sequence of *FOXM1* gene locus and recruiting the remodeling factor CBP to promote chromatin activation.

## CBP is also enriched in GSCs

As CBP is critical for regulating FOXM1 expression in GSCs, we next examined whether CBP is also preferentially expressed by GSCs. Immunoblot analyses showed that CBP was preferentially expressed in GSCs relative to matched NSTCs (Appendix Fig S5A) or NPCs (Appendix Fig S5B). The preferential expression of CBP in GSCs was further validated in primary human GBM samples. Immunofluorescent staining confirmed that CBP was highly expressed in nuclei of glioma cells expressing the GSC markers SOX2 and OLIG2 in primary human GBMs (Appendix Fig S5C and D). These data demonstrate that CBP is also preferentially expressed by GSCs in GBM.

## Inhibition of SATB2/CBP transcriptional activity by the CBP inhibitor C646 suppresses GSC-driven tumor growth

As SATB2/CBP plays critical roles in promoting GSC proliferation and GBM growth, we next explore whether pharmacologic inhibition of CBP activity by a small molecule inhibitor named C646 (Bowers *et al*, 2010) could impact GSC-driven tumor growth. When GSCs were treated with different concentrations of C646, GSC viability was markedly reduced in a dose-dependent manner (Fig 7A; Appendix Fig S6A). Consistently, C646 treatment significantly disrupted GSC tumorsphere formation in a dose-dependent manner (Fig 7B; Appendix Fig S6B–D). Further analysis showed that C646 treatment dramatically reduced expression of *FOXM1* and its target genes in a dose-dependent manner while had no effect on *SATB2* expression in GSCs (Fig 7C; Appendix Fig S6E). Immunoblot analysis confirmed that C646 treatment reduced FOXM1 protein levels in a dose-dependent manner (Appendix Fig S6F). To exclude the off-target effects of C646 treatment, we silenced the CBP expression by using CBP shRNA (shCBP) and then treated the GSCs with C646

(8 μM) or the vehicle control. Targeting CBP with shCBP significantly decreased CBP expression in GSCs (Appendix Fig S6G). We found that CBP knockdown attenuated the effect of C646 treatment on inhibiting GSC viability (Appendix Fig S6H). These results confirmed that CBP is the target of C646 on inhibiting GSC viability. To verify whether FOXM1 overexpression rescues the effect of C646 treatment on GSC maintenance, we transduced GSCs with FOXM1 and treated the GSCs with C646 (8 μM). Immunoblot analysis showed that ectopic expression of FOXM1 restored the FOXM1 levels to its endogenous levels that were reduced by C646 treatment (Appendix Fig S6I). Indeed, ectopic expression of FOXM1 to its endogenous levels largely rescued the impaired viability of GSCs caused by C646 treatment (Appendix Fig S6J). These results indicate that C646 treatment inhibits GSC viability through the CBP-FOXM1 axis. We then evaluated the sensitivity of NPCs to C646 and found that C646 treatment had little impact on NPC growth at the same concentration with GSCs (Appendix Fig S7A), indicating that targeting CBP by C646 may specifically affect GSCs. To compare the effects of C646 or temozolomide treatment alone and the combined treatment on GSC proliferation, we treated the GSCs with C646 (4 μM), temozolomide (40 μM), or in combination. The results showed that the inhibitory effect of C646 (4 μM) was slightly greater than that of temozolomide (40 μM) on GSC proliferation (Appendix Fig S7B), but the combined treatment with C646 (4 μM) and temozolomide (40 μM) showed a synergistic effect on inhibiting GSC proliferation (Appendix Fig S7B). To further compare the effect of C646 or irradiation treatment alone and the combined therapy on GSC proliferation, we treated GSCs with C646 (4 μM), irradiation (1 Gy), or in combination. The results showed that treatment with C646 (4 μM) or irradiation (1 Gy) alone had the similar effect on inhibiting GSC proliferation (Appendix Fig S7C). However, the combined treatment with C646 (4 μM) and irradiation (1 Gy) also showed a synergistic effect on inhibiting GSC proliferation (Appendix Fig S7C). These results indicate that targeting CBP with C646 may effectively synergize with the standard therapy such as irradiation or temozolomide treatment to improve GBM treatment.

As C646 has been shown to penetrate the blood-brain barrier (Baruch *et al*, 2015; Choi *et al*, 2017), we next determined the therapeutic impact of C646 treatment on GBM tumor growth in orthotopic xenograft models. Bioluminescent imaging demonstrated that C646 treatment significantly suppressed the GSC-driven tumor growth (Fig 7D–F; Appendix Fig S8A and B). As a consequence, C646 treatment significantly prolonged the survival of mice bearing GSC-derived tumors (Fig 7G; Appendix Fig S8C). Our preclinical study also indicated that C646 is well-tolerated, as little body weight loss was detected after the treatment (Appendix Fig S8D). In addition, C646 administration markedly reduced Ki67-positive proliferative cells in GSC-derived tumors (Fig 7H and I; Appendix Fig S8E and F). Thus, inhibiting the transcriptional activity of SATB2/CBP by the CBP inhibitor potently suppresses GBM tumor growth, suggesting that targeting this signaling axis may be a promising therapeutic strategy to effectively improve GBM treatment.

## Discussion

As one of key nuclear matrix-associated proteins (NMPs), SATB2 has been reported to regulate expression of certain genes during

development and cancer progression, but the role of SATB2 in glioma stem cells and GBM malignant growth has not been defined. In this study, we found that SATB2, a MAR-binding transcription factor, is crucial for maintaining GSC proliferation, self-renewal, and tumorigenic potential. We demonstrated that SATB2 promotes the GSC maintenance by binding to the MAR region of *FOXM1* gene and recruiting the histone acetyltransferase CBP to activate *FOXM1* expression (Fig 8). Moreover, genetic targeting or pharmacological inhibition of SATB2/CBP function significantly suppresses GSC-driven tumor growth (Fig 8). Our study indicates that the SATB2/CBP transcriptional complex plays critical roles in maintaining GSC property to sustain GBM tumor growth, suggesting that targeting the STAB2/CBP signaling axis may significantly improve survival of GBM patients.

We found that SATB2 mediates through FOXM1 to exert its function in GSCs. FOXM1 is a master transcription factor that promotes tumorigenesis and tumor progression mainly by stimulating expression of many genes involved in the cell cycle progression. Several studies have demonstrated that FOXM1 promotes malignant progression by activating the expression of cell cycle genes in various cancers such as liver cancer, breast cancer, and ovarian cancer (Yang *et al*, 2013; Barger *et al*, 2015; Hu *et al*, 2019). However, the molecular mechanisms underlying the FOXM1 regulation in cancer cells particularly in GSCs were poorly understood. In this study, we identified FOXM1 as a pivotal transcription target of the SATB2/CBP complex that promotes the proliferation and self-renewal of GSCs, although we cannot exclude the possibility that other factors may also partially contribute to the function of SATB2 in GSCs. Previous study reported that FOXM1 expression is higher in GSCs than in NPCs (Joshi *et al*, 2013). Consistently, our data show that SATB2 is highly expressed in GSCs relative to NPCs.

Previous studies have indicated that SATB2 may regulate gene transcription by binding to the MAR sequences of genomic DNA to modulate chromatin structure and function (Dickinson *et al*, 1997; Dobreva *et al*, 2003; Britanova *et al*, 2005; Yamasaki *et al*, 2007). SATB2 has been shown to promote Immunoglobulin μ expression by binding to MAR sequences of *Immunoglobulin* μ locus in pre-B cells (Dobreva *et al*, 2003). SATB2 is also expressed in erythroid cells and promotes *γ-Globin* gene expression by binding to MARs (Zhou *et al*, 2012). In addition, SATB2 regulates neural development by binding MAR sequences of *CTIP2* gene locus and repress *CTIP2* transcription (Britanova *et al*, 2008; Diaz-Alonso *et al*, 2012). However, the direct MAR-binding targets of SATB2 in cancers particularly in GBM have not been determined. Our study showed that SATB2 directly binds to the MAR sequence of *FOXM1* locus to promote its expression. We further demonstrated that SATB2 recruits histone acetyltransferase CBP to this MAR region to promote *FOXM1* transcription. When CBP is recruited by SATB2 to the MAR region, it can acetylate histone H3 and H4 at this region to regulate chromatin structure, as reflected by the increased Acetyl-Histone H3 (Lys18), Acetyl-Histone H3 (Lys27), and Acetyl-Histone H4. This binding has important functional significance because actively transcribed DNA is tightly associated with the nuclear matrix (Seo *et al*, 2005; Keaton *et al*, 2011). However, how the SATB2/CBP complex activates the MAR to modulate the gene transcriptional activity remains unclear and needs further investigation. MARs have been implicated to regulate gene transcription by altering the chromosome organization, defining the borders of chromatin domains and increasing the potential of enhancers to act over large distances (Forrester *et al*, 1994; Arope *et al*, 2013). Future study will clarify how the SATB2/CBP-activated MAR on *FOXM1* locus regulates transcriptional activity and expression levels of *FOXM1* in GSCs.

We further identified CBP as a binding partner of SATB2 in activating FOXM1 expression to promote GSC proliferation and GBM tumor growth. CBP is a histone acetyltransferase that regulates gene expression by acetylating histones or transcription factors (Ogryzko *et al*, 1996). CBP often serves as a transcriptional coactivator and has been shown to bind to a range of important transcription factors to promote downstream gene transcription (Wang *et al*, 2013). Recent studies have shown that CBP accelerates tumor growth in several cancers including colorectal cancer, colon cancer, lung cancer, and liver cancer (Wang *et al*, 2013; Xiao *et al*, 2015; Inagaki *et al*, 2016; Du *et al*, 2017; Qi & Zhao, 2019). However, the expression and function of CBP in GBMs were not determined. In this study, we demonstrated that CBP is preferentially expressed by GSCs in GBMs. Disrupting the SATB2/CBP complex significantly inhibited tumor growth by inhibiting FOXM1 expression, indicating that targeting the SATB2/CBP signaling axis may have therapeutic potential to improve GBM treatment. Importantly, we found that pharmacological inhibition of CBP activity significantly inhibited GSC proliferation and tumor growth in the mouse xenograft models, indicating that CBP is a promising therapeutic target for developing effective drugs to improve GBM treatment.

In summary, we identified SATB2/CBP as a critical regulatory complex to activate FOXM1 expression and promote the GSC maintenance as well as GBM malignant growth. SATB2/CBP binds to the MAR region of *FOXM1* gene locus to stimulate *FOXM1* transcription in GSCs. As pharmacological inhibition of SATB2/CBP function markedly inhibited GBM tumor growth, therapeutic targeting of the SATB2/CBP-FOXM1 signaling axis may offer an effective strategy to significantly improve therapeutic efficacy for GBMs and prolong survival of the patients.

# Materials and Methods

### Tissues, cells, and cell culture

Human GBM surgical tissues were obtained from patients at the Cleveland Clinic and University Hospitals of Case Western Reserve University for this study in accordance with an Institutional Review Board-approved protocol. Informed consent was obtained from all human subjects. All the experiments conformed to the principles set out in the WMA Declaration of Helsinki and the Department of Health and Human Services Belmont Report. GBM Specimens were verified by neuropathological examination. GSCs and matched NSTCs were isolated from primary GBM tumors or xenografts tumors through FACs as described (Bao *et al*, 2006; Guryanova *et al*, 2011; Zhou *et al*, 2017). In brief, cells were dissociated using Papain Dissociation Kit (Worthington Biochemical, Cat # LK003150) according to manufacturer's protocol. Dissociated cells were recovered in Neurobasal-A medium (Thermo Fisher, Cat # A2477501) supplemented with B27 (Thermo Fisher, Cat # 12587010), 10 ng/ml EGF(Gold biotechnology, Cat # 1150-04-1000), 10 ng/ml bFGF (R&D Systems, Cat # 4114-TC-01M), 2 mM L-glutamine (Thermo Fisher,

Cat # 35050061), and 1 mM sodium pyruvate (Thermo Fisher, Cat # 11360070) overnight. Cells were then labeled with a PE-conjugated anti-CD133 antibody (Miltenyi Biotec, Cat # 130-098-826, Clone AC133, 1:10) and a FITC-conjugated anti-CD15 antibody (BD Biosciences, Cat # 347423, Clone MMA, 1:10) followed by FACs to sort the GSCs (CD15$^+$/CD133$^+$) and NSTCs (CD15$^-$/CD133$^-$). GSCs were validated by the expression of GSC markers including SOX2, OLIG2, and L1CAM, self-renewal assay (serial tumorsphere formation), and tumor propagation assay (*in vivo* limiting dilution) (Bao *et al*, 2006; Guryanova *et al*, 2011; Fang *et al*, 2017). Matched NSTCs were cultured in DMEM with 10% FBS (Thermo Fisher, Cat # 10437028) to maintain differentiation status. Human neural progenitor cell lines (15167, 16157, NHNP, and NSC194) were derived from fetal brains and cultured in Neurobasal-A medium with supplements as described above (Guryanova *et al*, 2011; Fang *et al*, 2017).

### Orthotopic tumorigenesis and treatment

All animal procedures were approved by Cleveland Clinic Institutional Animal Care and Use Committee-approved protocols. Mice were housed at the Cleveland Clinic Lerner Research Institute Animal Care Facility and maintained at a temperature- and humidity-controlled environment with a 12-h light/12-h dark cycle. NSG mice (The Jackson Laboratory, Cat # 005557) used in this study were 6–10 weeks old. Intracranial xenografts were generated by implanting 5,000 GSCs into the right cerebral cortex of NSG mice at a depth of 3.5 mm. Animals were monitored by the bioluminescent imaging or maintained until neurological signs occurred. For the C646 (Selleckchem, Cat # S7152) treatment, 50 µl of 9 mg/kg C646 was dissolved in DMSO and was delivered once daily by intraperitoneal injection.

### Cell viability and *in vitro* limiting dilution assay

Cell viability assays were conducted by plating 1,000 cells (lentiviral infection) or 1,500 cells (drug treatment) per well in a 96-well plate, and cell number were measured using Cell Titer-Glo Luminescent Cell Viability Assay kit (Promega, Cat # G7571) according to the manufacturer's guidance. For the drug treatment, C646 (Selleckchem, Cat # S7152) or temozolomide (Sigma-Aldrich, Cat # T2577) were dissolved in DMSO and then were added to medium. For *in vitro* limiting dilution assay, indicated density of cells (0, 10, 20, 30, 40, 50 cells per well) with 30 replicates were seeded into one well of a 96-well pate. The presence and number of tumorspheres in each were recorded at day 6 after cell seeding, and the neurosphere formation efficiency was analyzed using the software at http://bioinf.wehi.edu.au/software/elda/.

### *In vivo* limiting dilution assay

5,000, 1,000, 500, or 100 GSCs were implanted into the right cerebral cortex of NSG mice at a depth of 3.5 mm through intracranial injection. Mice were maintained until the development of neurological signs or up to 12 weeks. Brains of euthanized mice were collected, fixed in 4% PFA, and embedded. Stem cell frequency was generated using the ELDA software (Hu & Smyth, 2009) at http://bioinf.wehi.edu.au/software/elda/.

### mRNA analysis, immunoblot analysis, and co-immunoprecipitation

Total RNA was isolated using PureLink RNA mini extraction kit (Thermo Fisher, Cat # 12183018A), reverse transcribed and analyzed by quantitative PCR using SYBR Green (Alkali Scientific, Cat # QS2050) and an ABI 7500 system (Applied Biosystems). Cycle threshold (Ct) values were defined as the cycle number at which the reporter fluorescence exceeded a fixed threshold. All results were expressed as Ct. Relative expression values are calculated using the $2^{-\Delta\Delta Ct}$ method (Livak & Schmittgen, 2001), where $\Delta C_t = Ct_{target\ gene} - Ct_{reference\ gene}$ and $\Delta\Delta C_T = \Delta CT_{sample} - \Delta CT_{calibrator}$. Using the $2^{-\Delta\Delta CT}$ method, data were shown as fold change in target gene expression, normalized to endogenous reference gene relative to the calibrator. A complete list of PCR primers is shown in Appendix Table S2. For immunoblot analysis, cells were lysed in RIPA buffer for 30 min and centrifuged at 15,000 *g* for 10 min. Protein samples were then separated by SDS–PAGE and transferred onto PVDF membranes. Blots were blocked with 5% non-fat milk for 1–2 h and then incubated with primary antibody overnight at 4°C followed by HRP-conjugated species-specific antibodies. Immunoreactivity was detected using BioRad Image Lab software. For co-immunoprecipitation, cells were lysed in IP lysis buffer for 30 min and centrifuged at 15,000 *g* for 10 min. Protein lysates were incubated with primary antibody and protein A/G Plus agarose beads (Santa Cruz, sc-2003, 20 µl) overnight at 4°C with constant rotation. The precipitants were washed with wash buffer for three times, boiled with SDS sample buffer, and subjected to immunoblot analysis. Primary antibodies listed as follows: SATB2 (Abcam, ab34735, 1:1,000 or Santa Cruz, sc-81376, Clone SATBA4B10, 1:200), SOX2 (Bethyl, A301-741A, 1:1,000), OLIG2 (Millipore, MABN50, clone 211F1.1, 1:1,000), CBP (Cell Signaling, 7389S, Clone D6C5, 1:1,000), P300 (Santa Cruz, sc-48343, Clone F-4, 1:300), GFAP (Biolegend, 644702, Clone 2E1.E9, 1:1,000), FOXM1 (Cell Signaling, 5436S, Clone D12D5, 1:500), and Tubulin (Sigma-Aldrich, 6199, Clone DM1A, 1:3,000).

### Plasmids and lentiviral transduction

Lentiviral vectors expressing two distinct shRNAs against human SATB2 (Cat # TRCN0000020685 or TRCN0000020688), human CBP (Cat # TRCN0000356053), and non-targeting shRNA (Cat # SHC002) were obtained from Sigma-Aldrich. A lentiviral construct expressing FOXM1 was generated by cloning the human FOXM1 open reading frame into the pCDH-EF1-MCs-IRES-Neo vector (System Biosciences, Cat # CD533A-2). The lentivirus packaging and transduction performed as previously described (Fang *et al*, 2017).

### In vivo bioluminescence analysis

GSCs expressing firefly luciferase were transduced with corresponding lentivirus. Forty-eight hours after lentiviral infection, 5,000 GSCs were intracranially transplanted into immunocompromised mice. To examine the tumor growth, animal brains were monitored by bioluminescent imaging at indicated days. Mice were injected with D-luciferin at 150 mg/kg intraperitoneally and then captured by Spectrum IVIS imaging system (PerkinElmer).

**The paper explained**

**Problem**

Glioblastoma (GBM) is the most frequent and malignant type of human primary brain tumor. The prognosis of GBM remains extremely poor despite significant advances in the treatment of other solid cancers. Accumulating evidence supports that glioma stem cells (GSCs) are responsible for GBM initiation, progression, and therapeutic resistance. Therefore, better understanding of the molecular mechanisms driving GSC proliferation and self-renewal may offer new therapeutic strategies targeting GSCs to inhibit GBM malignant growth and overcome the therapeutic resistance.

**Results**

Our study identified SATB2/CBP as a critical regulatory complex that activates FOXM1 expression in GSCs to promote the GSC maintenance and GBM malignant growth. Both SATB2 and CBP are preferentially expressed by GSCs in GBMs. SATB2/CBP binds to the matrix attachment region of *FOXM1* gene locus to stimulate *FOXM1* transcription in GSCs. Importantly, pharmacological inhibition of SATB2/CBP transcriptional activity by the CBP inhibitor C646 potently inhibited GBM tumor growth.

**Impact**

Our data demonstrate that targeting the SATB2/CBP-FOXM1 axis markedly inhibited GSC proliferation and GBM tumor growth, offering an effective therapeutic strategy through the inhibition of SATB2/CBP to improve GBM treatment.

## Immunofluorescent staining

Immunofluorescent staining of sections was performed as previously described (Zhou *et al*, 2017). Mouse brains bearing GBM xenografts were collected from mice when neurological signs occur after GSC transplantation and the tumors with similar size were selected for staining. Human primary GBM samples were collected from GBM patients through surgical resection. In brief, slides with PFA-fixed tumor tissues or cells were incubated with a PBS solution containing 1% BSA plus 0.3% Triton X-100 for 30 min, then incubated with diluted primary antibody at 4°C overnight. The slides were further incubated with the fluorescent second antibody for 2 h followed by DAPI for 5 min and then subjected to microscopy using an AMG EVOS FL microscope (Thermo Scientific) or a Leica DM4 B microscope (Leica). All staining were repeated three times. Staining on mice tissue samples was performed on sections from at least five tumors. Staining on human tissue samples was performed on sections from at least three GBM specimens. Primary antibodies listed as follows: SATB2 (Abcam, ab34735, 1:200; Santa Cruz, sc-81376, Clone SATBA4B10, 1:75; or Bethyl, A301-864A, 1:300), SOX2 (R&D, AF2018, 1:200 or Bethyl, A301-739A, 1:300), OLIG2 (R&D, AF2418, 1:100), CBP (Cell Signaling, 7389S, Clone D6C5, 1:100), Ki67 (Cell Signaling, 9129, Clone D3B5, 1:400), GFAP (Biolegend, 801103, Clone MCA-5C10, 1:100), TUBB3 (Biolegend, 801201,Clone TUJ1, 1:400), and GALC (Millipore, MAB342, clone mGalC, 1:300).

## EdU incorporation assay

For EdU incorporation assay, cells were incubated with Edu for 2h and then fixed with 4% PFA. The following staining was performed using the Click-iT Plus EdU Alexa Fluor Imaging Kit (Thermo Fisher, Cat # C10638) based on the manufacturer's protocol.

## Microarray and gene ontology analysis

Total RNA was isolated from GSCs expressing SATB2 shRNA or shNT. Gene expression profiles were performed at DNA Link, Inc. using the Affymetrix GeneChip Human Gene 2.0 ST Array. All procedures were carried out according to manufacturer's guideline. Gene ontology analysis was performed using the PANTHER classification system (http://pantherdb.org/).

## Chromatin immunoprecipitation assay

Chromatin immunoprecipitation was performed as described (Tao *et al*, 2011). In Brief, cells were fixed with 2.5% formaldehyde for 10 min. Chromatin lysates were prepared, pre-cleared with Protein A/G agarose beads, and immunoprecipitated with corresponding antibodies or control normal IgG in the presence of salmon sperm DNA and BSA. Beads were extensively washed and then reverse cross-linked at 65°C overnight. DNA was purified with a PCR purification kit (Qiagen, Cat # 28106) and subsequently analyzed by PCR or qPCR using primers flanking the MAR on the human *FOXM1* gene locus. MAR primers were as follows: forward primer, 5′-CTAGGC-GACAGAGCGAGACT-3′, and reverse primer, 5′-TTCCTGGCAGG-CAGTATTAAA-3′. Antibodies were as follows: SATB2 (Abcam, ab34735, 1:100), CBP (Cell Signaling, 7389S, Clone D6C5, 1:50), Acetyl-Histone H3 Lys18 (Cell Signaling, 9675S, 1:25), Acetyl-Histone H3 Lys27 (Cell Signaling, 8173S, Clone D5E4, 1:100), and Acetyl-Histone H4 (Thermo Fisher, PA1-84526, 1:100).

## Bioinformatic analysis

To identify the potential MARs, we analyzed a ~ 20 kb region upstream and downstream of the *FOXM1* gene's first exon to identify potential SATB2 binding by using Marscan tool (http://www.bioinformatics.nl/cgi-bin/emboss/marscan/). To determine the expression pattern of NMPs in patients with GBM or low-grade glioma, we interrogated TCGA database. The datasets generated were downloaded from TCGA portal (http://portal.gdc.cancer.gov/).

## Statistics

Data are presented as mean ± SD. For quantification with two groups, Mann–Whitney test was used to assess statistical significance with GraphPad Prism 7. For quantification with more than two groups, one-way ANOVA analysis followed by Tukey's test (Data with normal distribution) or Kruskal–Wallis test followed by Dunn's test (Data with non-normal distribution) was used to assess statistical significance with GraphPad Prism 7. For quantification with two or more groups and groups that have subgroups, two-way ANOVA analysis was used to assess statistical significance with GraphPad Prism 7. Data for two-way ANOVA analysis were normally distributed (Shapiro–Wilk normality test). Kaplan–Meier survival curves were generated using GraphPad Prism 7 and analyzed using the log-rank test, which is generally used for survival analysis (Cheng *et al*, 2013; Zhou *et al*, 2017). *P* value < 0.05 was considered statistically significant. Detailed

information is described in each figure panel. Exact *P* values are reported in Appendix Table S3. Randomization was used for animal studies. The studies were not performed blindly. Except for microarray experiment, similar results were obtained from three independent experiments.

## Data availability

The datasets produced in this study are available in the following databases:
- Microarray: Gene Expression Omnibus GSE154789 (http://www.ncbi.nlm.nih.gov/geo/query/acc.cgi?acc=GSE154789).

**Expanded View** for this article is available online.

## Acknowledgements

We thank the Brain Tumor and Neuro-Oncology Centers at Cleveland Clinic and the University Hospitals of Case Western Reserve University for providing GBM surgical specimens. We are very grateful for the help provided by Ms. Mary McGraw from the Brain Tumor Bank at Cleveland Clinic. We thank the Imaging Core and Flow Cytometry Core at Cleveland Clinic Lerner Research Institute for their assistance. This research was supported by grants from Cleveland Clinic Foundation and the NIH R01 grants (NS091080 and NS099175) to S. Bao, and the Peter D. Cristal Chair, the Kimble Family Foundation, and the Center of Excellence for Translational Neuro-Oncology at University Hospitals (A. E. Sloan). This work utilized the IVIS system (Spectrum CT) that was purchased with NIH Shared Instrumentation Grant (SIG) grant (S10OD018205).

## Author contributions

WT and SB designed the experiments. WT, AZ, KZ, ZH, HH, WZ, QH, XF, BCP, XW, and QW performed the experiments. AES and MSA provided some GBM surgical specimens. JDL, JSY, and JNR provided scientific input. WT and SB analyzed the data and prepared the manuscript.

## Conflict of interest

The authors declare that they have no conflict of interest.

## For more information

i  Marscan tool (http://www.bioinformatics.nl/cgi-bin/emboss/marscan/).
ii  TCGA portal (http://portal.gdc.cancer.gov/).
iii  PANTHER tool (http://pantherdb.org/).
iv  ELDA software (http://bioinf.wehi.edu.au/software/elda/).

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
