## [Review Process File · EMBO Molecular Medicine]

SATB2 Drives Glioblastoma Growth by Recruiting CBP to Promote FOXM1 Expression in Glioma Stem Cells

Weiwei Tao, Aili Zhang, Kui Zhai, Zhi Huang, Haidong Huang, Wenchao Zhou, Qian Huang, Xiaoguang Fang, Briana Prager, Xiuxing Wang, Qiulian Wu, Andrew Sloan, Manmeet Ahluwalia, Justin D. Lathia, Jennifer Yu, Jeremy Rich, and Shideng Bao

DOI: [10.15252/emmm.202012291](https://doi.org/10.15252/emmm.202012291)

Corresponding author: Shideng Bao (baos@ccf.org)

Review Timeline:

Submission Date:	4th Mar 20
Editorial Decision:	30th Apr 20
Revision Received:	12th Aug 20
Editorial Decision:	11th Sep 20
Revision Received:	19th Sep 20
Accepted:	22nd Sep 20

Editor: Zeljko Durdevic

Transaction Report:

30th Apr 2020

Dear Prof. Bao,

Thank you for the submission of your manuscript to EMBO Molecular Medicine, and please accept my apologies for the delay in getting back to you. We have now received feedback from two of the three reviewers who agreed to evaluate your manuscript. Given that referee #2 will unfortunately not be able to return his/her report in a timely manner, and that both referees #1 and #3 are overall positive, we prefer to make a decision now in order to avoid further delay in the process. Should referee #2 provide a report, we will send it to you, with the understanding that we will not ask you extensive experiments in addition to the ones required in the enclosed reports from referee #1 and #3. As you will see from the reports below, both referees highlight the interest of the study but also raise a number of concerns that should be addressed in a major revision of the current manuscript. Special attention should be given to addressing preclinical aspect of the study particularly comparing and combining the C646 treatment with standard GBM therapies. Addressing the reviewers' concerns in full will be necessary for further considering the manuscript in our journal.

Acceptance of the manuscript will entail a second round of review. Please note that EMBO Molecular Medicine encourages a single round of revision only and therefore, acceptance or rejection of the manuscript will depend on the completeness of your responses included in the next, final version of the manuscript. For this reason, and to save you from any frustrations in the end, I would strongly advise against returning an incomplete revision.

We realize that the current situation is exceptional on the account of the COVID-19/SARS-CoV-2 pandemic. Therefore, please let us know if you need more than three months to revise the manuscript.

I look forward to receiving your revised manuscript.

***** Reviewer's comments *****

Referee #1 (Remarks for Author):

Tao et al test a new molecular mechanism by which a key nuclear matrix-associated protein, SATB2, promotes GBM growth. The study is interesting and provides new information that is

important for our understanding of how NMPs regulate GBM progression. The authors identify SATB2 as marker of GSCs and poor prognosis in GBM patients. They then follow-up to report the mechanism of action of SATB2 and show that it recruits the histone acetyltransferase CBP to promote FOXM1-mediated cell proliferation. Finally, they use a CBP inhibitor that could be of utility for the field to block GBM cell growth in vitro and in vivo.

The work presented in the manuscript is valuable. However, there are some instances where the conclusions of the authors are not fully supported by the evidence.

Major points:

(1) The authors claim that 'SATB2 is required for GSC self-renewal'. This is solely based on in vitro assays, which is weak evidence. To strengthen this link, I would suggest including limiting dilution xenotransplantation of shSATB2 cells to demonstrate effects on tumorigenicity.

(2) The experiments using C646 to inhibit CBP function are interesting, but poorly controlled. No effort is made to exclude off-target effects of the inhibitor. The authors should use shCBP as control to show that this attenuates the effects of C646, and that C646 treatment effects can be rescued by over expression of FoxM1. It would also be useful to demonstrate that C646 treatment reduces protein levels of FoxM1.

(3) I have made several unsuccessful attempts to reproduce the Kaplan-Meier curves presented from the TCGA dataset on SATB2 and CBP. The authors need to provide better information which specific dataset and which cut-off parameters were used for the analysis.

(4) Statistical tests used should be better described, particularly where more than two groups were compared (e.g. all figure panels comparing shNT vs shSATB2-1 and shSATB2-2). Student's T-test is not appropriate for comparing 3 or more experimental groups. Was an attempt made to check if data is normally distributed? For sample numbers <30, a Mann-Whitney test is probably more appropriate than Student's T-test.

Minor points:

Median survival times should be presented for all Kaplan-Meier analyses.

Referee #3 (Remarks for Author):

The study by Tao et al. characterizes the role of SATB2, a nuclear matrix-associated protein (NMP) preferentially expressed by glioblastoma (GBM) stem-like cells (GSC), in sustaining GBM malignant growth in vitro and in vivo. The study provides a mechanistic explanation for the protumorigenic activity of SATB2, by uncovering that it recruits CBP transcription factor to activate transcription of FOXM1, a known regulator of cell cycle progression. Finally, the study provides evidence that pharmacological inhibition of SATB2/CBP transcriptional activity by a CBP inhibitor (C646) can suppress proliferation of GSC and growth of GBM obtained by GSC orthotopic transplantation. The study is ample, technically excellent, mostly carefully written and illustrated.

The findings concerning the SATB2/CBP/FOXM1 are novel and mechanistically appealing, as they investigate an area mostly unexplored in GBM but deserving attention, as it relates to chromatin regulation, known to be often altered during gliomagenesis.

A possible conceptual weakness is that SATB2, being a nuclear matrix-associated protein, can play

a 'machinery role' with elusive causal relationships with gliomagenesis. How can this protein be deregulated in GBM remains unclear: the alleged correlation between SATB2 high expression and poor prognosis is the less convincing result of the study (see Point N.1). Neither are clear the correlations between SATB2 and the GBM oncogenic pathways, and whether SATB2/CBP targeting with C646 has good chance to be GBM-specific in the patient.

The experiments are overall convincing, however, to fully support the main conclusions, the following points should be addressed.

Point N.1

Correlation between SATB2 high expression and poor prognosis. Results, page 6 and S1AC.

This reviewer challenged the association between SATB2 high expression and GBM prognosis in available GBM databases (using Gliovis and cbioportal), and found it hard to reproduce. The correlation seems negligible, except in the 'lower grade gliomas' cohort, where SATB2 expression directly correlates with glioma grade and, therefore, it inversely correlates with overall survival. If not adequately supported, such correlations should be removed from the results.

Point N.2

SATB2 preferential expression in GSCs. Results, page 6, Fig. 1A-H and S2A-C.

2A - In Fig 1A and S2A, 3 human GBM specimens are analysed. The number of non-GSC GBM cells (SOX2-neg) cells is very low according to DAPI-pos nuclei. This is unusual. Brightfield images should be shown.

2B - The conclusion that SATB2 is preferentially expressed in the GSC subpopulation vs. more differentiated non-stem tumor cells, is based on the comparison between GSCs and 'NSTC', i.e. cells where differentiation has been forced by 10% serum, which are unlikely representative of real GBM cells. To corroborate the above conclusion, it is suggested to investigate whether SATB2 expression is decreased in human GBM specimens, in cells showing expression of differentiation markers.

2C - The low levels of SATB2 in Neural Progenitor Cells (NPC) shown in Fig. 1F-G is puzzling, as NPC are known to express high levels of the SATB2 transcriptional target FOXM1 (see e.g.: Joshi et al. Stem Cells 31:1051, 2013).

Point N.3

SATB2 requirement for GSC proliferation and self-renewal. Results, page 7, Fig. S3.

3A - The statement that 'disruption of SATB2 had little effect on the growth and survival of NSTCs (Fig. S3A) and NPCs (Fig. S3B)' seems tautological, since, as shown in the previous result section, NSTCs and NPCs do not express SATB2. A similar problem is observed in Fig 6B, where ChIPs to study SATB2 binding are shown in NSTC or NPC.

3B - In Fig S3C the increased number of cells in G1 phase, and the decreased number in S, are statistically significant but biologically rather meaningless. This result should be confirmed in more independent models, or removed.

Point N. 4

Silencing SATB2 suppresses GSC-driven tumor growth. Results, page 7, Fig. 3A-G, S3E-G.

The overall study of experimental tumors should be better characterized and described. In particular:

4A. In the experimental tumors it is important to show the levels of SATB2. It is unclear at which timepoint the tumors shown in Fig. 3D-G were taken (the end-point?). Although 21 days after transplantation SATB2-silenced cells had not formed detectable tumors (Fig. 3A), in the following weeks tumors evidently appeared and were removed. It would be important to discriminate whether tumors arising from shSATB2 come from cells that escaped silencing (thus they should express

SATB2) or that kept their potential despite silencing (thus they should not express SATB2).
4B. Ki67 and SOX2 expression should be shown also in tumors shown in S3.

Point N. 5

SATB2 is required for the expression of genes involved in cell cycle progression. Results, page 8, Fig. 4A-G.

In SATB2 silenced cells, it is suggested to show not only targets that are upregulated, but also those that are downregulated by FOXM1, i.e. cell cycle inhibitors such as p21 and p27.

Point N. 6

SATB2 promotes GSC proliferation and tumor propagation through FOXM1, page 9, Fig. 5G-J.

As noted in point N.4, tumors should be better characterized, in particular for expression of SATB2, in order to understand what is the source of the tumor.

In this section, a previous study showing that FOXM1 is associated with GSC formation and regulates SOX2 expression should be cited (Lee et al. Plos One 10:e0137703).

Point N. 7

CBP is also enriched in GSCs, page 10, Fig. S5E.

As in Point N.1, this reviewer could not reproduce the alleged correlation between high CBP expression and poor prognosis. It is suggested to remove these data.

Point N. 8

The CBP inhibitor C646 suppresses GSC-driven tumor growth, page 11.

To provide valuable preclinical information, it would be important to compare and combine the effect of C646 with standard GBM therapies (temozolomide and radiotherapy), at least in vitro. This seems relevant also because it is known that FOXM1 plays an important role in regulating DNA damage repair machinery (the majority of its target genes are involved in response to IR and TMZ). For temozolomide treatment, it is recommended to use doses representative of concentrations measured in the patients' CNS, i.e. <50 uM.

Point N. 9

Statistical data.

9A. Data are presented as mean {plus minus} SD or SEM. It is suggested to uniform these data.

9B. As in many experiments data are collected from few experimental points (e.g. cell viability or sphere formation), it seems more appropriate to use non-parametric tests or to control (and disclose in the Methods section) that normal distribution can be applied. GraphPad Prism assists to the purpose.

Minor point

Introduction, page 4. The sentence 'elevated expression of SATB2 correlates with poor patient survival' is repeated almost identical at the beginning of page 5.

Responses to Reviewers' Comments

Re: EMM-2020-12291

We thank the reviewers for the critical evaluation of our manuscript. We are very grateful for the insightful comments and helpful suggestions from the reviewers. In response to their comments, we have performed a large amount of additional experiments and extensively revised the manuscript to address the major concerns. We believe that the revised manuscript is significantly improved and strengthened. Below, we include the point-by-point response to the reviewers' comments.

Referee #1 (Remarks for Author):

Tao et al test a new molecular mechanism by which a key nuclear matrix-associated protein, SATB2, promotes GBM growth. The study is interesting and provides new information that is important for our understanding of how NMPs regulate GBM progression. The authors identify SATB2 as marker of GSCs and poor prognosis in GBM patients. They then follow-up to report the mechanism of action of SATB2 and show that it recruits the histone acetyltransferase CBP to promote FOXM1-mediated cell proliferation. Finally, they use a CBP inhibitor that could be of utility for the field to block GBM cell growth in vitro and in vivo.

The work presented in the manuscript is valuable. However, there are some instances where the conclusions of the authors are not fully supported by the evidence.

Response: We are grateful for the positive comments from the reviewer. Meanwhile, we appreciate the helpful suggestions provided by the reviewer. We have performed additional experiments to address the important concerns.

Major points:

Referee #1: *1. The authors claim that 'SATB2 is required for GSC self-renewal'. This is solely based on in vitro assays, which is weak evidence. To strengthen this link, I would suggest including limiting dilution xenotransplantation of shSATB2 cells to demonstrate effects on tumorigenicity.*

Response: We thank the reviewer for the suggestion. We have performed the suggested experiment and confirmed that silencing SATB2 reduced the tumorigenic potential of GSCs in an *in vivo* limiting dilution assay (Please see Table R1 below).

shRNA	GSC number	5000	1000	500	100
shNT	Incidence	5/5	5/5	5/5	5/5
	Median Survival	29	36	41	57
shSATB2-1	Incidence	5/5	5/5	3/5	0/5
	Median Survival	49**	62**	73**	---**
shSATB2-2	Incidence	5/5	5/5	3/5	1/5
	Median Survival	47**	58**	67**	---**

Table R1. *In vivo* limiting dilution assay for tumor formation of GSCs expressing shNT or shSATB2.

Tumor incidence and median survival time of mice after intracranial transplantation of 5000, 1000, 500 or 100 GSCs (T3359) expressing shNT, shSATB2-1 or shSATB2-2. $**p < 0.01$ with Log-rank analysis of survival curves for the same number of GSCs expressing shSATB2 relative to shNT control.

We have added the new data in Appendix Table S1 and described the result in our revised manuscript. Please see the 3rd Paragraph at Page 7, the 3th part in the “Results” section: “Further experiment demonstrated that silencing SATB2 reduced the tumorigenic potential of GSCs in an in vivo limiting dilution assay.”

We have also added the description of the “*In vivo* limiting dilution assay” in the “Materials and Methods” section in the revised manuscript. Please see Page 18.

Referee #1: 2. The experiments using C646 to inhibit CBP function are interesting, but poorly controlled. No effort is made exclude off-target effects of the inhibitor. The authors should use shCBP as control to show that this attenuates the effects of C646, and that C646 treatment effects can be rescued by over expression of FoxM1. It would also be useful to demonstrate that C646 treatment reduces protein levels of FoxM1.

Response: We thank the reviewer for the helpful suggestion and agree that necessary controls are critical.

(1) To exclude the off-target effects of the inhibitor C646, we silenced the CBP expression by using CBP shRNA (shCBP) and then treated the GSCs with C646 (8 μ M) or the vehicle control. Targeting CBP with shCBP significantly decreased CBP expression in GSCs (Please see Figure R1A below). We found that CBP knockdown attenuated the effect of C646 treatment on inhibiting GSC proliferation (Please see Figure R1B below). These new results further confirmed that CBP is the target of C646 on inhibiting GSC proliferation.

We have added the new data in Appendix Fig S6G and H and described the results in our revised manuscript. Please see the Line 12 of 3rd Paragraph at Page 11, the last part in the “Results” section: “To exclude the off-target effects of C646 treatment.....These results confirmed that CBP is the target of C646 on inhibiting GSC proliferation.”

(2) To verify whether FOXM1 overexpression rescues the effect of C646 treatment on GSC proliferation, we transduced GSCs with FOXM1 by a lentiviral vector and treated the GSCs with C646 (8 μ M). Immunoblot analysis showed that ectopic expression of FOXM1 restored the FOXM1 levels to its endogenous levels that were reduced by C646 treatment (Please see Figure R2A below). Indeed, ectopic expression of FOXM1 to its endogenous levels largely rescued the impaired cell proliferation of GSCs caused by C646 treatment (Please see Figure R2B below). These results indicate that C646 treatment inhibits GSC proliferation through the CBP-FOXM1 axis.

We have added the new data in Appendix Fig S6I and J and described the result in our revised manuscript. Please see the last line at Page 11, the last part in the “Results” section: “To verify whether FOXM1 overexpression rescues the effect of C646 treatment on GSC proliferation.....These results indicate that C646 treatment inhibits GSC proliferation through the CBP-FOXM1 axis.”

(3) To test whether C646 treatment reduces FoxM1 protein levels in GSCs, we performed immunoblot analysis and found that C646 treatment indeed reduced FOXM1 expression in a dose-dependent manner (Please see Figure R3 below).

Figure R3. Immunoblot analysis of FOXM1 expression in GSCs treated with indicated doses of C646 for 24 hours.

We have added the new data in Appendix Fig S6F and described the result in our revised manuscript. Please see the Line 11 of 3rd Paragraph at Page 11, the last part in the “Results” section: “Immunoblot analysis confirmed that C646 treatment reduced FOXM1 protein levels in a dose-dependent manner.”

Referee #1: 3. I have made several unsuccessful attempts to reproduce the Kaplan-Meier curves presented from the TCGA dataset on SATB2 and CBP. The authors need to provide better information which specific dataset and which cut-off parameters were used for the analysis.

Response: We appreciate the important concern. The original survival curve on SATB2 was extracted from TCGA GBM (Agilent-4502A) database and the CBP result was extracted from TCGA GBM (HG-U133A) database. The statistical significance of the survival correlation was identified through bioinformatics analysis. Although there is a correlation at certain dataset by using specific cut-off parameter, the overall correlation is low by analyzing all available GBM databases. We understand that it may be more appropriate to use a median cut-off. Reviewer 3 (Point 1) also raised this point and suggested us to remove these data. Therefore, we have removed these data from the results according to the reviewer’s suggestion.

Referee #1: 4. Statistical tests used should be better described, particularly where more than two groups were compared (e.g. all figure panels comparing shNT vs shSATB2-1 and shSATB2-2). Student’s T-test is not appropriate for comparing 3 or more experimental groups. Was an attempt made to check if data is normally distributed? For sample numbers <30, a Mann-Whitney test is probably more appropriate than Student’s T-test.

Response: We thank the reviewer for raising this important issue. Student’s T-test was used to compare two groups in multiple groups in our previous manuscript (e.g. shNT vs shSATB2-1 or shNT vs shSATB2-2). We agree with the reviewer’s point that Student’s T-test may not be appropriate for comparing this type of data. Therefore, we re-analyzed the statistical data according to the reviewer’s suggestions. For quantification with more than two groups, one-way ANOVA analysis followed by Tukey’s test (Data with normal distribution) or Kruskal–Wallis test followed by Dunn’s test (Data with non-normal distribution) was used to assess statistical significance with GraphPad Prism 7. For quantification with two groups, Mann-Whitney test was used to assess statistical significance with GraphPad Prism 7. We have also added this statement in the “Materials and Methods” section in the revised manuscript. Please see Page 21 in the “Statistics” section.

Referee #1: 5. Minor points: Median survival times should be presented for all Kaplan-Meier analyses.

Response: We are sorry for missing the important information. We have added the median survival times for all survival curves at the corresponding figure legends.

Referee #3 (Remarks for Author):

The study by Tao et al. characterizes the role of SATB2, a nuclear matrix-associated protein (NMP) preferentially expressed by glioblastoma (GBM) stem-like cells (GSC), in sustaining GBM malignant growth in vitro and in vivo. The study provides a mechanistic explanation for the protumorigenic activity of SATB2, by uncovering that it recruits CBP transcription factor to activate transcription of FOXM1, a known regulator of cell cycle progression. Finally, the study provides evidence that pharmacological inhibition of SATB2/CBP transcriptional activity by a CBP inhibitor (C646) can suppress proliferation of GSC and growth of GBM obtained by GSC orthotopic transplantation.

The study is ample, technically excellent, mostly carefully written and illustrated.

The findings concerning the SATB2/CBP/FOXM1 are novel and mechanistically appealing, as they investigate an area mostly unexplored in GBM but deserving attention, as it relates to chromatin regulation, known to be often altered during gliomagenesis.

A possible conceptual weakness is that SATB2, being a nuclear matrix-associated protein, can play a 'machinery role' with elusive causal relationships with gliomagenesis. How can this protein be deregulated in GBM remains unclear: the alleged correlation between SATB2 high expression and poor prognosis is the less convincing result of the study (see Point N.1). Neither are clear the correlations between SATB2 and the GBM oncogenic pathways, and whether SATB2/CBP targeting with C646 has good chance to be GBM-specific in the patient.

The experiments are overall convincing, however, to fully support the main conclusions, the following points should be addressed.

Response: We thank the reviewer for the positive comments. We appreciate the concerns and suggestions raised by the reviewer. In response to the comments, we have performed a large amount of additional experiments to address the reviewer's concerns. We believe that the current manuscript is significantly improved after extensive revision in response to the constructive suggestions.

Referee #3: Point N.1

Correlation between SATB2 high expression and poor prognosis. Results, page 6 and S1A-C.

This reviewer challenged the association between SATB2 high expression and GBM prognosis in available GBM databases (using Gliovis and cbiportal), and found it hard to reproduce. The correlation seems negligible, except in the 'lower grade gliomas' cohort, where SATB2 expression directly correlates with glioma grade and, therefore, it inversely correlates with overall survival. If not adequately supported, such correlations should be removed from the results.

Response: We appreciate the important concern. In the original survival curves presented in the previous manuscript, the statistical significance of the correlation was identified through bioinformatics analysis. Although there is a correlation at certain dataset by using specific parameter, the overall correlation is low by analyzing all available GBM database. We understand that it may be more appropriate to use a median cut-off. We agree with the reviewer's opinion. Therefore, we have removed these data from the results.

Referee #3: Point N.2

SATB2 preferential expression in GSCs. Results, page 6, Fig. 1A-H and S2A-C.

2A - In Fig 1A and S2A, 3 human GBM specimens are analysed. The number of non-GSC GBM cells (SOX2-neg) cells is very low according to DAPI-pos nuclei. This is unusual. Brightfield images should be shown.

2B - The conclusion that SATB2 is preferentially expressed in the GSC subpopulation vs. more differentiated non-stem tumor cells, is based on the comparison between GSCs and 'NSTC', i.e. cells where differentiation has been forced by 10% serum, which are unlikely representative of real GBM cells. To corroborate the above conclusion, it is suggested to investigate whether SATB2 expression is decreased in human GBM specimens, in cells showing expression of differentiation markers.

2C - The low levels of SATB2 in Neural Progenitor Cells (NPC) shown in Fig. 1F-G is puzzling, as NPC are known to express high levels of the SATB2 transcriptional target FOXM1 (see e.g.: Joshi et al. Stem Cells 31:1051, 2013).

Response: (2A) We thank the reviewer for raising the issue. The proportion of SOX2⁺ or OLIG2⁺ GSCs among the total cancer cells in different GBM tumor samples actually varied a lot based on our experience. It happened that we were using some GBM samples with a high proportion of GSCs. We are sorry that we didn't take the pictures of bright fields from those samples in our previous staining. To address the reviewer's concern, we repeated the immunofluorescent staining in CW1797 and CW1798 GBM samples, as well as in the newly added CCF-4321 GBM sample, which contains a low proportion of GSCs than CW1797 and CW1798 samples. The results confirmed that SATB2 is indeed enriched in GSCs in primary GBM samples (Please see Figure R4 below). The bright field images are also shown in these figures (Figure R4 below).

human GBM specimens. Scale Bar, 25 μ M.

We have replaced the original Fig. 1A with this new figure.

(2B) To further confirm that SATB2 is preferentially expressed by GSCs relative to non-stem tumor cells (NSTCs), we performed immunofluorescent staining in human primary GBM samples using antibodies against SATB2 and three differentiation markers (GFAP, TUBB3 and GALC). We found that the majority of the NSTCs expressing the differentiation maker didn't show SATB2 signals (Please see Figure R5A-F below). Only small fractions of GFAP⁺ cells (7.8%), TUBB3⁺ cells (10.8%), and GALC⁺ cells (6.1%) showed SATB2 staining signals in human GBMs (Figure R5B, D and F), but the majority of SOX2⁺ cells (94%) or OLIG2⁺ cells (90.9%) showed strong SATB2 staining in human primary GBMs (Fig. 1B). These results further supports that SATB2 is preferentially expressed by GSCs in GBM tumors.

Figure R5. A, Immunofluorescence of SATB2 (red) and the astrocyte marker GFAP (green) in human GBM specimen. Scale Bar, 25 μ M.

B, Quantification of the fraction of SATB2⁺ cells in GFAP⁺ cells in human GBMs. Analysis were performed with 3 different specimens. Data are shown as mean \pm SD.

C, Immunofluorescence of SATB2 (red) and the neuron marker TUBB3 (green) in human GBM specimen. Scale Bar, 25 μ M.

D, Quantification of the fraction of SATB2⁺ cells in TUBB3⁺ cells in human GBMs. Analysis were

performed with 3 different specimens. Data are shown as mean \pm SD.

E, Immunofluorescence of SATB2 (red) and the oligodendrocyte marker GALC (green) in human GBM specimen. Scale Bar, 25 μ M.

F, Quantification of the fraction of SATB2⁺ cells in GALC⁺ cells in human GBMs. Analysis were performed with 3 different specimens. Data are shown as mean \pm SD.

We have added the new data in Appendix Fig S2B-G and described the result in our revised manuscript. Please see the Line 15 at Page 6, the 1st part in the “Results” section: “Further experiments demonstrated that SATB2 is rarely expressed in glioma cells expressing the differentiation markers (GFAP, TUBB3 and GALC) in human GBMs.”

(2C) We thank the reviewer for raising the interesting point. We have read the mentioned publication. The paper reported that FOXM1 expression was higher in GSCs than in NPCs, although FOXM1 was shown to be expressed by NPCs. This result is consistent with our data showing that SATB2 is highly expressed in GSCs compared with NPCs. On the other hand, it is possible that the regulatory mechanisms of FOXM1 expression may be different in GSCs and NPCs. Our results demonstrate that SATB2 is an important regulator of FOXM1 in GSCs. However, whether FOXM1 is also regulated by SATB2 in NPCs is not clear. The regulation of FOXM1 expression in NPCs and GSCs may not be same. We will further address this issue in the future study. We have discussed the issue in the “Discussion” section (Page 14, the 2nd Paragraph in the “Discussion” section) and cited the publication (Joshi et al. Stem Cells, 2013) in our manuscript.

Referee #3: Point N.3

SATB2 requirement for GSC proliferation and self-renewal. Results, page 7, Fig. S3.

3A - The statement that 'disruption of SATB2 had little effect on the growth and survival of NSTCs (Fig. S3A) and NPCs (Fig. S3B)' seems tautological, since, as shown in the previous result section, NSTCs and NPCs do not express SATB2. A similar problem is observed in Fig 6B, where chIPs to study STAB2 binding are shown in NSTC or NPC.

3B - In Fig S3C the increased number of cells in G1 phase, and the decreased number in S, are statistically significant but biologically rather meaningless. This result should be confirmed in more independent models, or removed.

Response: (3A) We thank the reviewer for raising the issue. Our results showed that NSTCs and NPCs still express very low levels of SATB2. Therefore, we performed the experiments. We believe that these experiments can serve as controls to demonstrate that SATB2 disruption specifically affects GSCs.

(3B) We agree with the reviewer that the increased number of cells in G1 phase and the decreased number in S phase shown in original Fig. S3C are not biologically meaningful. Thus, we have removed this data.

Referee #3: Point N. 4

Silencing SATB2 suppresses GSC-driven tumor growth. Results, page 7, Fig. 3A-G, S3E-G.

The overall study of experimental tumors should be better characterized and described. In particular:

4A. In the experimental tumors it is important to show the levels of SATB2. It is unclear at which timepoint the tumors shown in Fig. 3D-G were taken (the end-point?). Although 21 days after transplantation SATB2-silenced cells had not formed detectable tumors (Fig. 3A), in the following weeks tumors evidently appeared and were removed. It would be important to discriminate whether tumors arising from shSATB2 come from cells that escaped silencing (thus they should express SATB2) or that kept their potential despite silencing (thus they should not express SATB2).

4B. Ki67 and SOX2 expression should be shown also in tumors shown in S3.

Response: (4A) We appreciate the concern and suggestion raised by the reviewer. Because the SATB2-silenced GSCs grew tumor much slower than the control GSCs expressing shNT, it was very hard to collect tumors derived from the SATB2-silenced GSCs and the control GSCs at the same time point for immunofluorescence analyses. To make tumors from all groups comparable, we collected tumors from all groups at the similar tumor size for the analyses based on similar bioluminescence intensity. Thus, we usually collected the brains bearing GBM xenografts from mice two to three days before the death of mice, and the tumor size was determined by IVIS imaging. Although we collected tumors from the control group (shNT) and shSATB2 groups at different times after transplantation, tumor sizes from these three groups (shNT, shSATB2-1 and shSATB2-2) were similar and comparable for the immunofluorescence analyses. We have described the collection time of tumor xenografts in the “Materials and Methods-Immunofluorescent staining” section (Page 20) in our revised manuscript.

We agree with the reviewer that it is important to determine whether tumors arising from shSATB2 came from the GSCs that escaped silencing or that kept their potential despite silencing. To address this issue, we performed immunofluorescent staining of SATB2 in GBM xenografts and found that most of the cells in the GBM tumors derived from the shSATB2-cells didn't show SATB2 signals, although there were small fractions of cells in these tumors showed very weak SATB2 signals (Please see Figure R6A and B below). This result demonstrated that the glioma cells in GBM xenografts from the shSATB2 groups were indeed derived from SATB2-silenced cells but not derived from the GSCs that escaped from shRNA silencing.

expressing shNT or shSATB2 (n=5 tumors per group). Scale bar: 40 μ m.

B, Quantification of SATB2 intensity in xenografts derived from T3359 GSCs expressing shNT or shSATB2 (n=5 tumors per group). Data are shown as mean \pm SD. **** p <0.0001, one way ANOVA analysis followed by Tukey's test.

We have added the new data in Fig 3H and I and described the result in our revised manuscript. Please see the Line 4 at Page 8, the 3rd part in the "Results" section: "Moreover, immunofluorescent staining confirmed that the expression of SATB2 was significantly decreased....."

(4B) Following the reviewer's suggestion, we have also performed immunofluorescent staining of Ki67, SOX2 and SATB2 in GSC-derived GBM xenografts. The results showed that silencing SATB2 significantly reduced cell proliferation as revealed by Ki67 immunofluorescence and reduced the GSC population marked by SOX2 immunofluorescence (Please see Figure R7A-D below). Immunofluorescent staining of SATB2 also indicated that most of the cells in the GBM tumors from the shSATB2 groups didn't show SATB2 signals, and there were small fractions of cells in these tumors showed very weak SATB2 signals (Please see Figure R7E and F below).

Figure R7. A, Immunofluorescence of Ki67 (Green) in tumor xenografts derived from H2S GSCs expressing shNT or shSATB2 (n=5 tumors per group). Scale bar: 40 μ m.

B, Quantification of Ki67 positive cells in xenografts derived from H2S GSCs expressing shNT or shSATB2 (n=5 tumors per group). Data are shown as mean \pm SD. **** p <0.0001, one way ANOVA analysis followed by Tukey's test.

C, Immunofluorescence of SOX2 (Red) in xenografts derived from H2S GSCs expressing shNT or shSATB2 (n=5 tumors per group). Scale bar: 40 μ m.

D, Quantification of SOX2 positive cells in xenografts derived from H2S GSCs expressing shNT or shSATB2 (n=5 tumors per group). Data are shown as mean \pm SD. **** p <0.0001, one way ANOVA analysis followed by Tukey's test.

E, Immunofluorescence of SATB2 (Green) in GBM xenografts derived from H2S GSCs expressing shNT or shSATB2 (n=5 tumors per group). Scale bar: 40 μ m.

F, Quantification of SATB2 intensity in xenografts derived from H2S GSCs expressing shNT or shSATB2 (n=5 tumors per group). Data are shown as mean \pm SD. **** p <0.0001, one way ANOVA analysis followed by Tukey's test.

We have added the new data in Appendix Fig S3G-L and in result part of our revised manuscript at Page 8.

Referee #3: Point N. 5

SATB2 is required for the expression of genes involved in cell cycle progression. Results, page 8, Fig. 4A-G.

In SATB2 silenced cells, it is suggested to show not only targets that are upregulated, but also those that are downregulated by FOXM1, i.e. cell cycle inhibitors such as p21 and p27.

Response: We thank the reviewer for the suggestion. We have examined the expression of p21 and p27. qPCR analysis showed that knockdown of SATB2 slightly increased the expression of p27, while significantly increased p21 expression in GSCs (Please see Figure R8 below).

Figure R8. qPCR analysis of p21 and p27 expression in H2S GSCs transduced with shNT or shSATB2. Data are represented as mean \pm SD. * p <0.05, ** p <0.01, one way ANOVA analysis followed by Tukey's test.

We have added the new data in Fig 4G and described the result in our revised manuscript. Please see the Line 1 at Page 9, the 4th part in the “Results” section: “while disruption of SATB2 increased the expression of p21 and p27 which are negatively regulated by FOXM1.”

Referee #3: Point N. 6

SATB2 promotes GSC proliferation and tumor propagation through FOXM1, page 9, Fig. 5G-J.

As noted in point N.4, tumors should be better characterized, in particular for expression of SATB2, in order to understand what is the source of the tumor.

In this section, a previous study showing that FOXM1 is associated with GSC formation and regulates SOX2 expression should be cited (Lee et al. Plos One 10:e0137703).

Response: (6A) We thank the reviewer for the suggestion. We have performed immunofluorescent staining of SATB2 and confirmed that most of the cells in the GBM tumors expressing SATB2 shRNA didn't show SATB2 signals, and there were small fractions of cells in these tumors showed very weak SATB2 signals (Please see Figure R9 below). The result is consistent with the result shown in Figure R6, demonstrating that the glioma cells in GBM xenografts expressing SATB2 shRNA were indeed derived from SATB2-silenced cells but not derived from the GSCs that escaped from shRNA silencing.

We have added the new data in Appendix Fig S4E and F and described the result in our revised manuscript. Please see the Line 16 of 2nd Paragraph at Page 9, the 5th part in the “Results” section: “Immunofluorescent staining confirmed a significant reduction of SATB2 expression in xenografts expressing shSATB2.”

(6B) We are sorry for missing the important reference. We have cited it in the “Results” section. Please see the Line 2 of 2nd Paragraph at Page 9, the 5th part in the “Results” section: “As FOXM1 is an

oncogenic regulator that promotes GSC proliferation and expression of the stem cell marker SOX2 (Lee et al., 2015).”

Referee #3: Point N. 7

CBP is also enriched in GSCs, page 10, Fig. S5E.

As in Point N.1, this reviewer could not reproduce the alleged correlation between high CBP expression and poor prognosis. It is suggested to remove these data.

Response: We agree with the reviewer’s opinion. Although there is a correlation in certain dataset by using specific parameter, the overall correlation is low by analyzing all available GBM database. We have removed the data from the results.

Referee #3: Point N. 8

The CBP inhibitor C646 suppresses GSC-driven tumor growth, page 11.

To provide valuable preclinical information, it would be important to compare and combine the effect of C646 with standard GBM therapies (temozolomide and radiotherapy), at least in vitro. This seems relevant also because it is known that FOXM1 plays an important role in regulating DNA damage repair machinery (the majority of its target genes are involved in response to IR and TMZ). For temozolomide treatment, it is recommended to use doses representative of concentrations measured in the patients’ CNS, i.e. <50 μ M.

Response: We thank the reviewer for the important suggestion. To compare the effects of C646 and Temozolomide treatment alone and the combined treatment on GSC proliferation, we treated the GSCs with C646 (4 μ M) or Temozolomide (40 μ M) or in combination. The results showed that the inhibitory effect of C646 (4 μ M) was slightly greater than that of Temozolomide (40 μ M) on GSC proliferation (Please see Figure R10 below), but the combined treatment with C646 (4 μ M) and Temozolomide (40 μ M) showed a synergistic effect on inhibiting GSC proliferation (Figure R10).

Figure R10. Cell viability assay of GSCs treated with C646 (4 μ M) or Temozolomide (40 μ M) or in combination for 6 days (n=5). Data are represented as mean \pm SD. **** p <0.0001, one way ANOVA analysis followed by Tukey’s test.

To further compare the effect of C646 and irradiation treatment alone and the combined treatment on GSC proliferation, we treated the GSCs with C646 (4 μ M) or irradiation (1 Gy) or in combination. The results showed that treatment with C646 (4 μ M) alone and with irradiation (1 Gy) alone had the similar effect on inhibiting GSC proliferation (Please see Figure R11 below). However, the combined treatment with C646 (4 μ M) and irradiation (1 Gy) also showed a synergistic effect on inhibiting GSC proliferation (Figure R11).

These results indicate that targeting CBP with C646 may effectively synergize with standard therapies such as irradiation or Temozolomide treatment to improve GBM treatment. We have added these new data in Appendix Fig S7B and C and described the results in our revised manuscript. Please see the Line 9 of Page 12, the last part in the "Results" section: "To compare the effects of C646 and Temozolomide treatment alone and the combined treatment on GSC proliferation.....These results indicate that targeting CBP with C646 may effectively synergize with standard therapies such as irradiation or Temozolomide treatment to improve GBM treatment "

Referee #3: Point N. 9

Statistical data.

9A. Data are presented as mean {plus minus} SD or SEM. It is suggested to uniform these data.

9B. As in many experiments data are collected from few experimental points (e.g. cell viability or sphere formation), it seems more appropriate to use non-parametric tests or to control (and disclose in the Methods section) that normal distribution can be applied. GraphPad Prism assists to the purpose.

Response: (9A) We have uniformed the data. All data are represented as mean \pm SD now.

(9B) We thank the reviewer for raising the important issue. We re-analyzed the statistical data according to the reviewer's suggestions. For quantification with two groups, Mann-Whitney test (non-parametric test) was used to assess statistical significance with GraphPad Prism 7. For quantification with more than two groups, one-way ANOVA analysis followed by Tukey's test (Data with normal distribution) or Kruskal–Wallis test followed by Dunn's test (Data with non-normal distribution) was used to assess statistical significance with GraphPad Prism 7. For quantification with two or more groups and groups that have subgroups, two-way ANOVA analysis was used to assess statistical significance with GraphPad Prism 7. Data for two-way ANOVA analysis were normally distributed (Shapiro-Wilk normality test). We have added this statement in the "Materials and Methods" section in the revised manuscript. Please see Page 21 in the "Statistics" section.

Referee #3: Minor point

Introduction, page 4. The sentence 'elevated expression of SATB2 correlates with poor patient survival' is repeated almost identical at the beginning of page 5.

Response: We have removed this sentence as we removed the correlation data. We thank the reviewer for the insightful comments that helps us to improve the manuscript.

11th Sep 2020

Dear Prof. Bao,

Thank you for the submission of your revised manuscript to EMBO Molecular Medicine. We have now received the enclosed reports from the referees that were asked to re-assess it. As you will see the reviewers are now globally supportive and I am pleased to inform you that we will be able to accept your manuscript pending the following final amendments:

***** Reviewer's comments *****

Referee #1 (Remarks for Author):

The authors have successfully addressed all my comments. There are only a couple minor points that should be tweaked for added clarity.

1) Authors have successfully performed the in vivo limiting dilution assay confirming that silencing SATB2 reduced the tumorigenic potential of GSCs. However, they should include stem cell frequency among groups (using ELDA analysis) and not only medium survival. This should be explained in the methodology section.

2) In Table R1 shCBP reduces cell proliferation but in the figure legend this is described as cell viability. Cell viability and cell proliferation are two different cell characteristics.

Referee #3 (Comments on Novelty/Model System for Author):

I confirm my comments to the first version of the manuscript:

Technical quality/adequacy of model system: The in vitro and in vivo models are adequate and the experiments have been carefully conducted, and satisfactorily and clearly described.

The findings concerning the SATB2/CBP/FOXO1 are novel and mechanistically appealing, as they investigate an area mostly unexplored in GBM but deserving attention, as it relates to chromatin regulation, known to be often altered during gliomagenesis.

Medical impact: it is unclear how SATB2 can be altered in glioblastomas; however, an inhibitor is available to further assess the targeting of this pathway in preclinical models.

Referee #3 (Remarks for Author):

All the points I raised have been adequately addressed. I congratulate for the high quality of the study and the revision.

Please further consider these points:

- 1- Fig. 1A and S2A-G: which kind of 'GBM specimens' are represented? They seem primary cell cultures. Please specify in the figure legend or in the text.
- 2- In methods, please add the mRNA quantification methodology leading to the 'relative mRNA level' represented in several graphs.
- 3- The result presented in Fig. 3H, showing the persistence of SATB2 silencing after sh transduction, should be presented at the beginning and not at the end of the paragraph.
- 4- Discussion, page 14. The added citation is appropriate, however the conclusion 'indicating that FOXM1 may also be regulated by SATB2 in NPCs', in the absence of experimental evidence seems overstated, and should be better removed.

Responses to Reviewers' Comments**Referee #1** (Remarks for Author):

The authors have successfully addressed all my comments. There are only a couple minor points that should be tweaked for added clarity.

Response: We thank the reviewer for the positive comment and the additional suggestions.

1) Authors have successfully performed the in vivo limiting dilution assay confirming that silencing SATB2 reduced the tumorigenic potential of GSCs. However, they should include stem cell frequency among groups (using ELDA analysis) and not only medium survival. This should be explained in the methodology section.

Response: We have included the stem cell frequency in the result (Appendix Table S1). We also added the description of the generation of stem cell frequency in the "Materials and Methods- In vivo limiting dilution assay" section (Page 18) of our revised manuscript.

2) In Table R1 shCBP reduces cell proliferation but in the figure legend this is described as cell viability. Cell viability and cell proliferation are two different cell characteristics.

Response: We apologized for the inappropriate description of the result. We used the cell viability assay in this experiment. Therefore, we changed the description of "cell proliferation" to "cell viability" in the result section of our revised manuscript. Please see the Lines 15-17 of 3rd Paragraph at Page 11.

Referee #3 (Comments on Novelty/Model System for Author):

I confirm my comments to the first version of the manuscript:

Techincal quality/adequacy of model system: The in vitro and in vivo models are adequate and the experiments have been carefully conducted, and satisfactorily and clearly described.

The findings concerning the SATB2/CBP/FOX M1 are novel and mechanistically appealing, as they investigate an area mostly unexplored in GBM but deserving attention, as it relates to chromatin regulation, known to be often altered during gliomagenesis.

Medical impact: it is unclear how SATB2 can be altered in glioblastomas; however, an inhibitor is available to further assess the targeting of this pathway in preclinical models.

Response: We thank the reviewer for the positive comments and the effort to review our manuscript. The reviewer is right. We still don't know how SATB2 is altered in glioblastoma. We will address this point in the future.

Referee #3 (Remarks for Author):

All the points I raised have been adequately addressed. I congratulate for the high quality of the study and the revision.

Response: We are very grateful for the positive comment from the reviewer.

Please further consider these points:

1- Fig. 1A and S2A-G: which kind of 'GBM specimens' are represented? They seem primary cell cultures. Please specify in the figure legend or in the text.

Response: We used the frozen tissue sections of human GBM surgical specimens to perform immunofluorescent staining in these figures. We have modified the previous description to "Immunofluorescence ofon frozen tissue sections of human GBM surgical specimens" in these figure legends.

2- In methods, please add the mRNA quantification methodology leading to the 'relative mRNA level' represented in several graphs.

Response: We have added the methodology in the "Materials and methods" section of our revised manuscript. Please see the Line 3 of 1st Paragraph at Page 19.

3- The result presented in Fig. 3H, showing the persistence of SATB2 silencing after sh transduction, should be presented at the beginning and not at the end of the paragraph.

Response: We have amended this part. Please see the Line 13 of 3rd Paragraph at Page 7. Figures were also moved forward accordingly.

4- Discussion, page 14. The added citation is appropriate, however the conclusion 'indicating that FOXM1 may also be regulated by SATB2 in NPCs', in the absence of experimental evidence seems overstated, and should be better removed.

Response: We thank the reviewer for raising the point. We have removed this sentence.

The authors performed the requested changes.

Corresponding Author Name: Shideng Bao
Journal Submitted to: EMBO Molecular Medicine
Manuscript Number: EMM-2020-12291-V2